# DIFF-2-IN-1: BRIDGING GENERATION AND DENSE PERCEPTION WITH DIFFUSION MODELS

**Shuhong Zheng**[1] **Zhipeng Bao**[2] **Ruoyu Zhao**[3] **Martial Hebert**[2] **Yu-Xiong Wang**[1]
[1]University of Illinois Urbana-Champaign, [2]Carnegie Mellon University, [3]Tsinghua University
{szheng36, yxw}@illinois.edu {zbao, hebert}@cs.cmu.edu
zhao-ry20@mails.tsinghua.edu.cn

## ABSTRACT

Beyond high-fidelity image synthesis, diffusion models have recently exhibited promising results in dense visual perception tasks. However, most existing work treats diffusion models as a standalone component for perception tasks, employing them solely for off-the-shelf data augmentation or as mere feature extractors. In contrast to these isolated and thus sub-optimal efforts, we introduce an integrated, versatile, diffusion-based framework, Diff-2-in-1, that can simultaneously handle both multi-modal data generation and dense visual perception, through a unique exploitation of the *diffusion-denoising process*. Within this framework, we further enhance discriminative visual perception via multi-modal generation, by utilizing the denoising network to create multi-modal data that mirror the distribution of the original training set. Importantly, Diff-2-in-1 optimizes the utilization of the created diverse and faithful data by leveraging a novel self-improving learning mechanism. Comprehensive experimental evaluations validate the effectiveness of our framework, showcasing consistent performance improvements across various discriminative backbones and high-quality multi-modal data generation characterized by both realism and usefulness. Our project website is available at https://zsh2000.github.io/diff-2-in-1.github.io/.

## 1 INTRODUCTION

Diffusion models have emerged as powerful tools for generating high-fidelity images Song et al. (2021); Ho et al. (2020); Rombach et al. (2022); Zhang et al. (2023b). Beyond their primary synthesis capabilities, diffusion models are increasingly recognized for their expressive representation abilities. This has spurred interest in leveraging them for dense visual perception, such as semantic segmentation (Baranchuk et al., 2022; Wu et al., 2023; Xu et al., 2023a) and depth estimation (Saxena et al., 2023b; Zhao et al., 2023). Nonetheless, most existing approaches treat diffusion models as a *standalone* component for perception tasks, either employing them for standard data augmentation (Burg et al., 2023) or utilizing the diffusion network as a feature extractor (Xu et al., 2023a; Zhao et al., 2023; Ji et al., 2023; Saxena et al., 2023a). These efforts overlook the *unique* diffusion-denoising process inherent in diffusion models, thus limiting their potential for discriminative tasks.

Inspired by foundational studies that explore the interplay between generative and discriminative learning (Rubinstein & Hastie, 1997; Ng & Jordan, 2001; Raina et al., 2003; Ulusoy & Bishop, 2005), we argue that the diffusion-denoising process plays a critical role in unleashing the capability of diffusion models for the discriminative visual perception tasks. The diffusion process corrupts the visual input with noise, enabling the *generation* of abundant new data with diversity. Subsequently, the denoising process removes the noise from noisy images to create high-fidelity data, thus obtaining informative features for *discriminative* tasks at the same time. As a result, the diffusion-denoising process naturally connects the generative process with discriminative learning.

Interestingly, this synergy further motivates us to propose a novel *integrated* diffusion modeling framework capable of performing discriminative and generative learning. From the generative perspective, we focus on synthesizing photo-realistic *multi-modal* paired data (*i.e.*, RGB images and their corresponding pixel-level visual attributes) that accurately capture various types of visual information. Meanwhile, the integrated diffusion model can achieve promising results in different

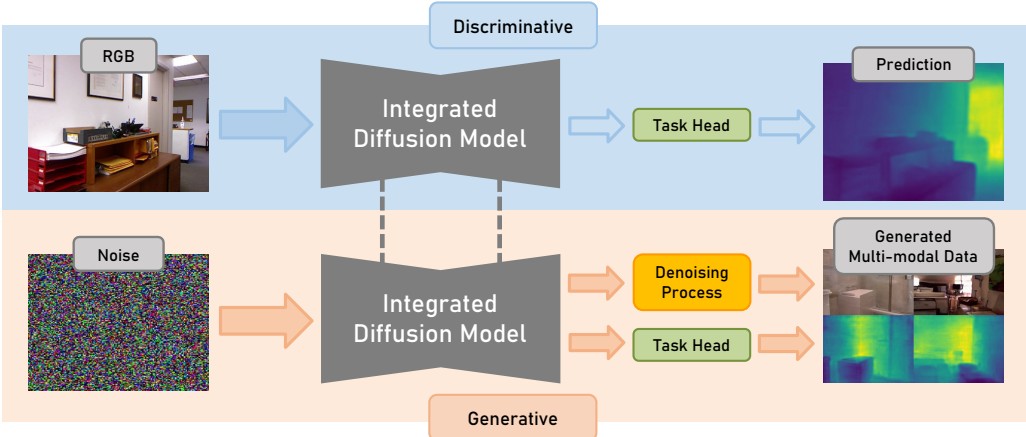

Figure 1: A single, integrated diffusion-based model capable of performing both multi-modal generation and dense perception. When the model receives an RGB image as input, it predicts an accurate visual attribute map. Simultaneously, the model is equipped to produce photo-realistic and coherent multi-modal data sampled from Gaussian noise.

visual prediction tasks from the discriminative standpoint. As an example illustrated in Figure 1, when considering RGB and depth interactions, when the model receives an RGB image as input, it predicts an accurate depth map. Meanwhile, the model can produce photo-realistic and coherent RGB-depth pairs sampled from noise. Despite its conceptual simplicity, fully implementing the integrated framework to improve both multi-modal generation and dense perception – such as effectively leveraging generated samples for discriminative tasks – remains challenging. In particular, the generation process inevitably produces data of relatively inferior quality compared to real data and may exhibit considerable data distribution gaps from the target domain.

To address these challenges, we introduce Diff-2-in-1, a diffusion framework bridging multi-modal generation and discriminative dense visual perception within one integrated diffusion model. The core design within our Diff-2-in-1 is a self-improving learning mechanism, featuring two sets of parameters for our integrated diffusion model *during the training process*. Specifically, the *creation parameters* are tailored to generate additional multi-modal data for discriminative learning, while the *exploitation parameters* are employed for utilizing both the original and synthetic data to learn the discriminative dense visual perception task. Meanwhile, the creation parameters continuously undergo *self-improvement* based on the weights of the exploitation parameters via exponential moving average (EMA). With our novel design of two sets of parameters interplaying with each other, the discriminative learning process can benefit from the synthetic samples generated by the model itself, while the quality of the generated data is iteratively refined at the same time.

We validate the effectiveness of Diff-2-in-1 through extensive and multi-faceted experimental evaluations. We start with the evaluation of the discriminative perspective, demonstrating its superiority over state-of-the-art discriminative baselines across various tasks in both single-task and multi-task settings. We additionally show that Diff-2-in-1 is generally applicable to different backbones and consistently boosts performance. Next, we ablate the experimental settings such as different training data sizes, to gain a comprehensive understanding of our method. Finally, we demonstrate the realism and usefulness of the multi-modal data generated by our Diff-2-in-1.

Our contributions include: **(1)** We propose Diff-2-in-1, an integrated framework that seamlessly performs multi-modal generation and dense visual perception based on diffusion models. **(2)** We introduce a novel self-improving mechanism that progressively enhances multi-modal generation in a self-directed manner, thereby effectively boosting the discriminative performance via generative learning. **(3)** Our method demonstrates consistent performance improvements across various discriminative backbones and high-quality multi-modal generation under both realism and usefulness.

## 2 RELATED WORK

**Generative modeling for discriminative tasks.** The primary objective of generative models has traditionally been synthesizing photo-realistic images. However, recent advancements have expanded their utility to the generation of "useful" images for downstream visual tasks (Zhan et al., 2018; Zhu et al., 2018; Aleotti et al., 2018; Pilzer et al., 2018; Zhang et al., 2023c; Zhu et al., 2024; Zheng et al., 2023b; Bao et al., 2022). This is typically accomplished by generating images and corresponding annotations off-the-shelf, subsequently using them for data augmentation in specific visual tasks.

Nowadays, with the emergence of powerful diffusion models in high-fidelity synthesis tasks (Song et al., 2021; Ho et al., 2020; Rombach et al., 2022; Zhang et al., 2023b; Wang et al., 2022; Chen et al., 2023), there has been a growing interest in applying them to discriminative tasks. Among them, ODISE (Xu et al., 2023a) and VPD (Zhao et al., 2023) extract features using the stable diffusion model (Rombach et al., 2022) to perform discriminative tasks such as segmentation and depth estimation. DIFT (Tang et al., 2023) and its concurrent work (Luo et al., 2023; Zhang et al., 2023a; Hedlin et al., 2023) utilize diffusion features for identifying semantic correspondence. DDVM (Saxena et al., 2023a) solves depth and optical flow estimation tasks by denoising from Gaussian noise with RGB images as a condition. Diffusion Classifier (Li et al., 2023a) utilizes diffusion models to enhance the confidence of zero-shot image classification. Another line of research including Marigold (Ke et al., 2024), Hyperhuman (Liu et al., 2024a), GeoWizard (Fu et al., 2024), StableNormal (Ye et al., 2024) repurposes diffusion models from image generation to dense prediction by finetuning the denoising network. With such a design, they achieve promising results with the cost of totally losing the capability of generation. In comparison, we exploit the capability of diffusion models to discriminative perception, and at the same time, preserve the original RGB generation capability, further expanding to multi-modal generation. Other studies (Trabucco et al., 2024; Feng et al., 2023; Burg et al., 2023) have explored using diffusion models to augment training data for classification. Different from them, we propose an *integrated* diffusion-based model that can directly perform dense visual perception, and simultaneously utilize its multi-modal generation capability to facilitate discriminative learning through the proposed novel self-improving algorithm.

## 3 INTEGRATED DIFFUSION MODEL: DIFF-2-IN-1

### 3.1 PRELIMINARY: LATENT DIFFUSION MODELS

Diffusion models (Ho et al., 2020) are latent variable models that learn the data distribution with the inverse of a Markov noising process. We build our method upon latent diffusion model (LDM) (Rombach et al., 2022). First, an encoder $\mathcal{E}$ is trained to map an input image $x \in \mathcal{X}$ into a spatial latent code $z = \mathcal{E}(x)$. A decoder $\mathcal{D}$ is then tasked with reconstructing the input image such that $\mathcal{D}(\mathcal{E}(x)) \approx x$. To convert a clean latent $z_0$ to a noisy latent $z_T$ of arbitrary timestep $T$, we have:

$$z_T \sim q(z_T | z_0) = \mathcal{N}(z_T; \sqrt{\bar{\alpha}_T} z_0, (1 - \bar{\alpha}_T)\mathbf{I}), \tag{1}$$

where the notation $\alpha_T = 1 - \beta_T$ and $\bar{\alpha}_T = \prod_{s=1}^{T} \alpha_s$ makes the formulation concise, $\beta_T$ controls the strength of the noise added in timestep $T$. When $T \to \infty$, $z_T$ is nearly equivalent to sampling from an isotropic Gaussian distribution.

The denoising process takes inverse operations from the diffusion process. We estimate the denoised latent at timestep $t - 1$ from $t$ by:

$$p_\theta(z_{t-1} | z_t) = \mathcal{N}(z_{t-1}; \mu_\theta(z_t, t), \boldsymbol{\Sigma}_\theta(z_t, t)), \tag{2}$$

where the parameters $\mu_\theta(z_t, t), \boldsymbol{\Sigma}_\theta(z_t, t)$ of the Gaussian distribution are estimated from the model.

### 3.2 AN INTEGRATED MODEL BEYOND RGB GENERATION

In this section, we use diffusion-based models for both discriminative and generative tasks to form our Diff-2-in-1 framework. Concretely, for a diffusion-based integrated model $\Phi$, we want it to predict task label $\hat{y} = \Phi^{\text{dis}}(x)$ given input image $x$; meanwhile, after training, it can generate multi-modal paired data from Gaussian: $(\tilde{x}, \tilde{y}) = \Phi^{\text{gen}}(\epsilon)$. We describe how we achieve this below.

**Discriminative perspective.** Previous work (Xu et al., 2023a; Zhao et al., 2023) has demonstrated the possibility of using diffusion models for perceptual tasks. Following VPD (Zhao et al., 2023),

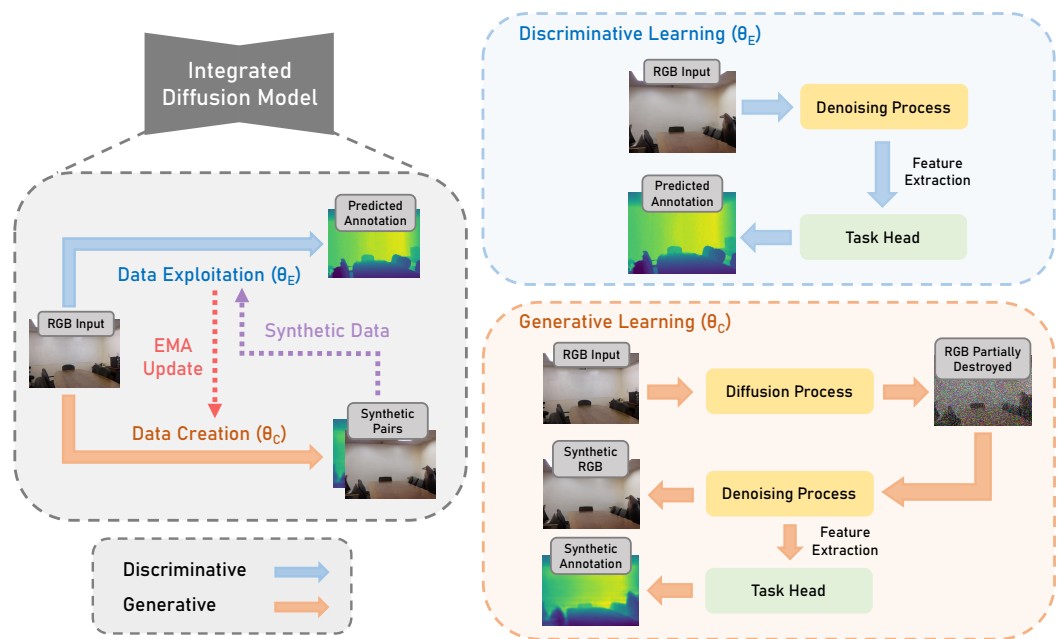

Figure 2: Our self-improving learning paradigm with two sets of interplayed parameters. The data creation parameter $\theta_C$ generates additional training samples for the data exploitation parameter $\theta_E$, while $\theta_E$ performs discriminative learning and provides guidance to update $\theta_C$ through exponential moving average. Finally, $\theta_C$ performs both discriminative and generative tasks during inference.

with the latent code $z = \mathcal{E}(x)$ from given image $x$, we perform one-step denoising on $z$ through the denoising U-Net (Ronneberger et al., 2015) to produce multi-scale features. Afterward, we rescale and concatenate those features and further pass them to a task head for downstream prediction.

**Generative perspective.** To generate paired multi-modal data, we first produce a latent vector $\tilde{z}_0$ by denoising from Gaussian with conditional text. Next, we directly generate the color image $\tilde{x}$ by passing it to the LDM decoder; meanwhile, we perform another one-step denoising with $\tilde{z}_0$ and send the resulting multi-scale features to the task head to obtain the corresponding label $\tilde{y}$.

The two perspectives reflect different usages of the integrated diffusion model while they are *not* fully separated: performing generation can be treated as a process of denoise-and-predict for a noisy image at timestep $t = T$; predicting labels can be treated as a process of data generation conditioned on a given latent vector $z_0$. This special connection motivates the design of our Diff-2-in-1.

## 4 LEARNING MECHANISM OF DIFF-2-IN-1

To effectively leverage the generated multi-modal data for dense visual perception, we propose a *self-improving* mechanism for our Diff-2-in-1 framework to make the discriminative and generative processes interact with each other, as shown in Figure 2. The details are described as below.

### 4.1 WARM-UP STAGE

Since pretrained diffusion models are only designed for RGB generation, we need a warm-up stage to activate the task head in Figure 2 for additional tasks. To achieve this, we train our integrated diffusion model using its discriminative learning pipeline with all the original training data with loss

$$\mathcal{L} = \sum_{i=1}^{N} \mathcal{L}_{\text{sup}}(f_{\theta_W}(\boldsymbol{x}_i), \boldsymbol{y}_i), \tag{3}$$

where $\mathcal{L}_{\text{sup}}$ is the supervised loss for our chosen discriminative task on the original paired training data $D_{\text{train}} = \{\boldsymbol{x}_i, \boldsymbol{y}_i\}_{i=1}^{N}$. We obtain a set of parameter weights $\theta_W$ after this warm-up stage.

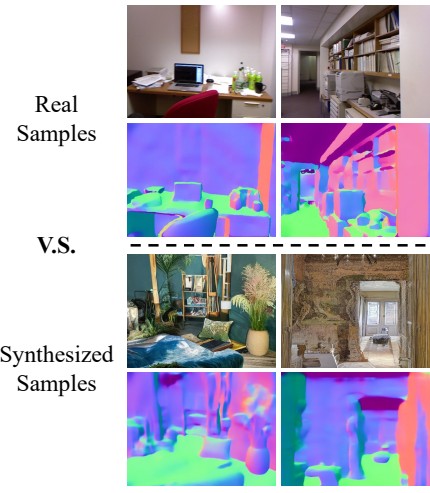

Real Samples

**V.S.**

Synthesized Samples

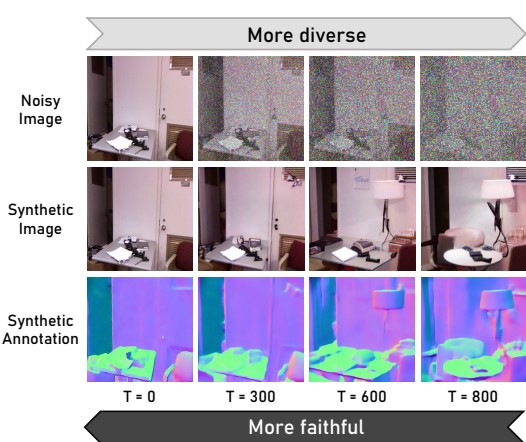

Figure 3: Real data samples from NYUv2 and synthesized samples generated from Gaussian. The distribution of the generated data varies from real data.

Figure 4: In-distribution data generation using partial noise. When denoising from a noisy image at timestep $T$ with $0 < T < T_{\max}$, a larger $T$ leads to greater diversity, whereas a smaller $T$ enhances the resemblance to the original distribution.

## 4.2 DATA GENERATION

Many approaches (Feng et al., 2023; Burg et al., 2023) that use diffusion models for data augmentation generate data from Gaussian noise as discussed in Section 3.2. However, as shown in Figure 3, the synthetic samples generated from Gaussian noise have a non-negligible distribution shift from the original training data, posing huge obstacles to utilizing the generated data for boosting the discriminative task performance. To narrow down the domain gap between the generated data and original data, inspired by SDEdit (Meng et al., 2022) and DA-Fusion (Trabucco et al., 2024), we use the inherent diffusion-denoising mechanism to control the data generation process.

Concretely, we add noise to the latent $z_i$ of an image $\boldsymbol{x}_i$ from the training set using Equation 1 at a timestep $T$ satisfying $0 < T < T_{\max}$, where $T_{\max}$ is the maximum timestep in the training process of diffusion models ($T_{\max} = 1000$ for all our experiments). This process partially corrupts the image with noise, yet maintains a degree of the original content, as depicted in the first row of Figure 4. After denoising the noisy image with Equation 2 and decoding with the variational autoencoder, we obtain the synthetic image $\tilde{\boldsymbol{x}}_i$ with different content but a relatively small domain gap, as shown in the second row of Figure 4. At the same time, we can obtain the prediction $\tilde{\boldsymbol{y}}_i$ which is decoded from the task head of the integrated diffusion model. As shown in the third row of Figure 4, the generated annotations (surface normal as an example) well match the generated RGB images. The timestep $T$, representing the noise level, acts as a modulator, balancing the diversity of the generated samples and the fidelity to the in-distribution data: higher noise levels lead to greater diversity, whereas lower levels enhance the resemblance to the original distribution.

## 4.3 SELF-IMPROVING STAGE

While synthetic multi-modal data typically demonstrates high visual fidelity, its direct utility for discriminative learning remains uncertain. To more effectively utilize the generated multi-modal data, we propose a self-improving mechanism inspired by the mean teacher learning system (Tarvainen & Valpola, 2017). As shown in Figure 2, our self-improving mechanism introduces the following two sets of parameters, both are initialized with $\theta_{\mathrm{W}}$, to iteratively perform the self-improvement for both generative and discriminative learning. The functions of these two sets of parameters are elaborated as follows.

**Data creation network** ($\theta_{\mathrm{C}}$) is used to create samples through the generative process within our integrated diffusion model. During every iteration, for a batch of $m$ real paired data $\{(\boldsymbol{x}_i, \boldsymbol{y}_i)\}_{i=1}^{m}$, we additionally generate $n$ paired samples $\{(\tilde{\boldsymbol{x}}_i, \tilde{\boldsymbol{y}}_i)\}_{i=1}^{n}$ with $\theta_{\mathrm{C}}$ following the data creation scheme described in Section 4.2. Both real and synthetic data are used for data exploitation.

| Model | Training Samples | 11.25° (↑) | 22.5° (↑) | 30° (↑) | Mean (↓) | Median (↓) | RMSE (↓) |
|---|---|---|---|---|---|---|---|
| SkipNet (Bansal et al., 2016) | 795 | 47.9 | 70.0 | 77.8 | 19.8 | 12.0 | 28.2 |
| GeoNet (Qi et al., 2018) | 30,816 | 48.4 | 71.5 | 79.5 | 19.0 | 11.8 | 26.9 |
| PAP (Zhang et al., 2019) | 12,795 | 48.8 | 72.2 | 79.8 | 18.6 | 11.7 | 25.5 |
| GeoNet++ (Qi et al., 2022) | 30,816 | 50.2 | 73.2 | 80.7 | 18.5 | 11.2 | 26.7 |
| Bae et al. (2021) | 30,816 | 62.2 | 79.3 | 85.2 | 14.9 | 7.5 | 23.5 |
| Bae et al. (2021) | 795 | 56.6 | 76.8 | 83.0 | 17.2 | 9.3 | 26.6 |
| GNA on Bae *et al.* | 795 | 56.4 | 76.7 | 83.0 | 17.3 | 9.3 | 26.7 |
| DA-Fusion (Trabucco et al., 2024) on Bae *et al.* | 795 | 58.1 | 77.5 | 83.6 | 16.8 | 8.9 | 26.1 |
| Diff-2-in-1 on Bae *et al.* (Ours) | 795 | 67.4 | 83.4 | 88.2 | 13.2 | 6.5 | 22.0 |
| iDisc (Piccinelli et al., 2023) | 30,816 | 63.8 | 79.8 | 85.6 | 14.6 | 7.3 | 22.8 |
| iDisc (Piccinelli et al., 2023) | 795 | 57.3 | 76.4 | 82.9 | 17.8 | 8.8 | 26.4 |
| GNA on iDisc | 795 | 56.9 | 76.2 | 82.4 | 18.1 | 8.9 | 26.7 |
| DA-Fusion (Trabucco et al., 2024) on iDisc | 795 | 58.7 | 78.3 | 83.4 | 17.3 | 8.6 | 26.2 |
| Diff-2-in-1 on iDisc (Ours) | 795 | **68.7** | **83.7** | **88.4** | **12.7** | **6.0** | **21.6** |

Table 1: Surface normal evaluation on NYUv2 (Silberman et al., 2012; Ladicky et al., 2014). When applying our Diff-2-in-1 on top of state-of-the-art baselines, we achieve consistently and significantly better performance with notably fewer training data, demonstrating the advantages of data efficiency from our integrated diffusion model. Additionally, Diff-2-in-1 outperforms augmentation methods GNA and DA-Fusion, proving the usefulness of the multi-modal data generated by our pipeline, and the effectiveness of our self-improving mechanism in utilizing synthetic data.

**Data exploitation network ($\theta_{\mathbf{E}}$)** is used for exploring the parameter space by exploiting both the original and the synthetic data samples to learn the discriminative task. With those $m + n$ samples, $\theta_{\mathrm{E}}$ is updated via the discriminative loss:

$$\mathcal{L} = \sum_{i=1}^{m} \mathcal{L}_{\mathrm{sup}}(f_{\theta_{\mathrm{E}}}(\boldsymbol{x}_i), \boldsymbol{y}_i) + \sum_{i=1}^{n} \mathcal{L}_{\mathrm{syn}}(f_{\theta_{\mathrm{E}}}(\tilde{\boldsymbol{x}}_i), \tilde{\boldsymbol{y}}_i), \tag{4}$$

where $\mathcal{L}_{\mathrm{syn}}$ is the loss term for synthetic data for which we regard the generated annotation $\tilde{\boldsymbol{y}}_i$ as the ground truth. It has the same format as the supervised loss $\mathcal{L}_{\mathrm{sup}}$.

**Feedback from data exploitation: EMA optimization.** The additional generated data from $\theta_{\mathrm{C}}$ facilitate the discriminative learning of $\theta_{\mathrm{E}}$. To further promote the interaction of the two sets of parameters within the integrated diffusion model, $\theta_{\mathrm{E}}$ provides $\theta_{\mathrm{C}}$ with gradient guidance as the feedback in response via the exponential moving average (EMA) strategy:

$$\theta_{\mathrm{C}} \leftarrow \alpha \theta_{\mathrm{C}} + (1 - \alpha) \theta_{\mathrm{E}}, \tag{5}$$

where $\alpha \in [0, 1)$ is a momentum hyperparameter that is usually set to close to 1. A large $\alpha$ maintains the overall quality of the generated data, preventing $\theta_{\mathrm{C}}$ from getting distracted by the inevitable inferior data. With the feedback from $\theta_{\mathrm{E}}$ to $\theta_{\mathrm{C}}$, the generated multi-modal data further get refined, in turn providing higher-quality and more reliable data back to $\theta_{\mathrm{E}}$ to achieve a *self-improving* cycle.

After the self-improvement, only one set of parameter, $\theta_{\mathrm{C}}$, is used to perform both generative and discriminative tasks during inference. The parameters in the diffusion model are kept frozen in default settings unless otherwise specified, allowing more flexibility and lightweight finetuning with less burden for the computational overhead.

## 5 EXPERIMENTAL EVALUATION

### 5.1 EVALUATION SETUP

We first evaluate our proposed Diff-2-in-1 in the single-task settings with surface normal estimation and semantic segmentation as targets. Next, we apply Diff-2-in-1 in multi-task settings of NYUD-MT (Silberman et al., 2012) and PASCAL-Context (Mottaghi et al., 2014) to show that it can provide universal benefit for more tasks simultaneously.

**Datasets and metrics.** We evaluate surface normal estimation on the **NYUv2** (Silberman et al., 2012; Ladicky et al., 2014) dataset. Different from previous methods that leverage additional raw data for training, we only use the 795 training samples. We include the number of training samples for each method in Table 1 for reference. Following Bae et al. (2021) and iDisc (Piccinelli et al., 2023), we adopt $11.25°, 22.5°, 30°$ to measure the percentage of pixels with lower angle error than the corresponding thresholds. We also report the mean/median angle error and the root mean square

| Model | mIoU (↑) |
|---|---|
| Swin-L (Liu et al., 2021b) | 52.1 |
| ConvNeXt-L (Liu et al., 2022) | 53.2 |
| ConvNeXt-XL (Liu et al., 2022) | 53.6 |
| MAE-ViT-L/16 (He et al., 2022) | 53.6 |
| CLIP-ViT-B (Rao et al., 2022) | 50.6 |
| VPD (Zhao et al., 2023) | 53.7 |
| DA-Fusion (Trabucco et al., 2024) on VPD | 54.0 |
| Diff-2-in-1 on VPD (Ours) | **54.5** |

Table 2: Semantic segmentation evaluations on ADE20K (Zhou et al., 2017) dataset. We compare with the diffusion-based segmentation method VPD (Zhao et al., 2023) and its baseline models. Our proposed Diff-2-in-1 further improves the performance of the diffusion-based VPD model.

| Model | Semseg mIoU (↑) | Depth RMSE (↓) | Normal mErr (↓) |
|---|---|---|---|
| Cross-stitch (Misra et al., 2016) | 36.34 | 0.6290 | 20.88 |
| PAP (Zhang et al., 2019) | 36.72 | 0.6178 | 20.82 |
| PSD (Zhou et al., 2020) | 36.69 | 0.6246 | 20.87 |
| PAD-Net (Xu et al., 2018) | 36.61 | 0.6270 | 20.85 |
| NDDR-CNN (Gao et al., 2019) | 36.72 | 0.6288 | 20.89 |
| MTI-Net (Vandenhende et al., 2020) | 45.97 | 0.5365 | 20.27 |
| ATRC (Bruggemann et al., 2021) | 46.33 | 0.5363 | 20.18 |
| DeMT (Xu et al., 2023c) | 51.50 | 0.5474 | 20.02 |
| MQTransformer (Xu et al., 2023b) | 49.18 | 0.5785 | 20.81 |
| DeMT (Xu et al., 2023c) | 51.50 | 0.5474 | 20.02 |
| InvPT (Ye & Xu, 2022) | 53.56 | 0.5183 | 19.04 |
| DA-Fusion (Trabucco et al., 2024) on InvPT | 53.70 | 0.5167 | 18.81 |
| Diff-2-in-1 on InvPT (Ours) | 54.71 | **0.5015** | 18.60 |
| TaskPrompter (Ye & Xu, 2023) | 55.30 | 0.5152 | 18.47 |
| DA-Fusion (Trabucco et al., 2024) on TaskPrompter | 55.13 | 0.5065 | 18.15 |
| Diff-2-in-1 on TaskPrompter (Ours) | **55.73** | 0.5041 | **17.91** |

Table 3: Comparison with state-of-the-art methods on the multi-task NYUD-MT (Silberman et al., 2012) benchmark. Our Diff-2-in-1 brings additional performance gains.

| Model | Semseg mIoU (↑) | Parsing mIoU (↑) | Saliency maxF (↑) | Normal mErr (↓) |
|---|---|---|---|---|
| ASTMT (Maninis et al., 2019) | 68.00 | 61.10 | 65.70 | 14.70 |
| PAD-Net (Xu et al., 2018) | 53.60 | 59.60 | 65.80 | 15.30 |
| MTI-Net (Vandenhende et al., 2020) | 61.70 | 60.18 | 84.78 | 14.73 |
| ATRC-ASPP (Bruggemann et al., 2021) | 63.60 | 60.23 | 83.91 | 14.30 |
| ATRC-BMTAS (Bruggemann et al., 2021) | 67.67 | 62.93 | 82.29 | 14.24 |
| MQTransformer (Xu et al., 2023b) | 71.25 | 60.11 | 84.05 | 14.74 |
| DeMT (Xu et al., 2023c) | 75.33 | 63.11 | 83.42 | 14.54 |
| InvPT (Ye & Xu, 2022) | 79.03 | 67.61 | 84.81 | 14.15 |
| DA-Fusion (Trabucco et al., 2024) on InvPT | 79.33 | 68.45 | 84.45 | 14.04 |
| Diff-2-in-1 on InvPT (Ours) | 80.36 | 69.55 | 84.64 | 13.89 |
| TaskPrompter (Ye & Xu, 2023) | 80.89 | 68.89 | **84.83** | 13.72 |
| DA-Fusion (Trabucco et al., 2024) on TaskPrompter | 80.81 | 69.23 | 84.47 | 13.70 |
| Diff-2-in-1 on TaskPrompter (Ours) | **80.93** | **69.73** | 84.35 | **13.64** |

Table 4: Comparison on the multi-task PASCAL-Context (Mottaghi et al., 2014) benchmark. Equipped with our Diff-2-in-1, the state-of-the-art methods reach an overall better performance.

error (RMSE) of all pixels. We evaluate semantic segmentation on the **ADE20K** (Zhou et al., 2017) dataset and use mean Intersection-over-Union (mIoU) as the metric. For multi-task evaluations, **NYUD-MT** spans across three tasks including semantic segmentation, monocular depth estimation, and surface normal estimation; **PASCAL-Context** takes semantic segmentation, human parsing, saliency detection, and surface normal estimation for evaluation. We adopt mIoU for semantic segmentation and human parsing, RMSE for monocular depth estimation, maximal F-measure (maxF) for saliency detection, and mean error (mErr) for surface normal estimation, following the same standard evaluation schemes (Misra et al., 2016; Zhang et al., 2019; Zhou et al., 2020; Xu et al., 2018; Gao et al., 2019; Vandenhende et al., 2020; Bruggemann et al., 2021; Xu et al., 2023b; Maninis et al., 2019; Xu et al., 2023c; Ye & Xu, 2022; 2023).

**Key implementation details.** To speed up training, instead of creating the paired data on the fly which takes significantly longer time due to denoising, we pre-synthesize a certain number of RGB images and later use $\theta_C$ to produce corresponding labels during the self-improving stage. More details about datasets, baselines, and implementations are included in Section A in the appendix.

## 5.2 DOWNSTREAM TASK EVALUATION

**Surface normal estimation.** We build our Diff-2-in-1 on two state-of-the-art surface normal prediction frameworks: Bae et al. (2021) and iDisc (Piccinelli et al., 2023). Our Diff-2-in-1 creates 500 synthetic pairs with timestep $T = 600$ (refer to Section 4.2). Besides conventional methods, we include two additional baselines with diffusion-based data augmentation. *DA-Fusion* (Trabucco et al., 2024) generates in-distribution RGB images with labels sharing a similar spirit as us, but only focuses on improving image classification task. To adapt it for dense pixel prediction, we adopt an off-the-shelf captioning strategy (Li et al., 2023b) to replace its textual inversion and apply the pretrained instantiated model to get the pixelwise annotations for the generated images. Afterward, the generated RGB-annotation pairs are utilized in the same way as DA-Fusion originally

| Model | $T$ | 11.25° (↑) | 22.5° (↑) | 30° (↑) | Mean (↓) | Median (↓) | RMSE (↓) |
|---|---|---|---|---|---|---|---|
| | 300 | 67.2 | 83.3 | 88.1 | 13.3 | 6.6 | 22.1 |
| Diff-2-in-1 on Bae et al. (2021) | 600 | **67.4** | **83.4** | **88.2** | **13.2** | **6.5** | **22.0** |
| | 800 | 67.3 | 83.3 | 88.1 | 13.3 | 6.6 | 22.1 |
| | 300 | 68.6 | 83.6 | **88.4** | 12.8 | **6.0** | **21.6** |
| Diff-2-in-1 on iDisc (Piccinelli et al., 2023) | 600 | **68.7** | **83.7** | **88.4** | **12.7** | **6.0** | **21.6** |
| | 800 | 68.5 | 83.6 | 88.3 | 12.8 | **6.0** | **21.6** |

Table 5: Ablation study on different timesteps $T$ during the data generation process within Diff-2-in-1. A medium timestep $T = 600$ achieves the best performance, but overall Diff-2-in-1 is robust to different choices of $T$.

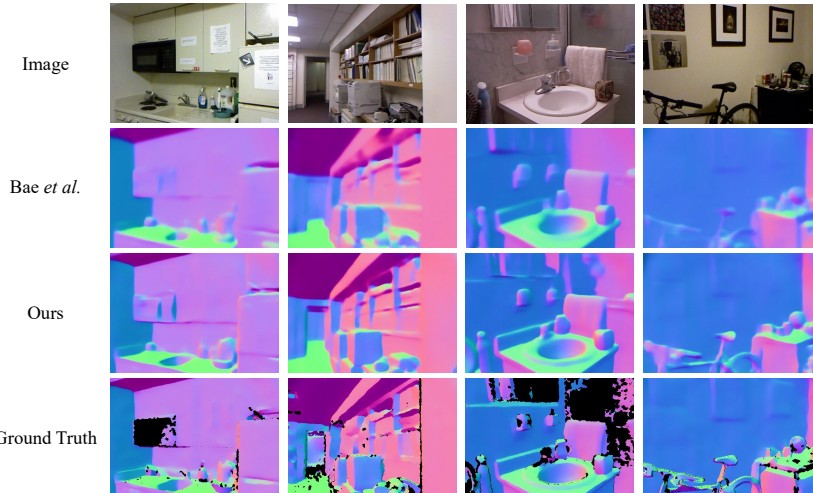

Figure 5: Qualitative results on the surface normal prediction task of NYUv2 (Silberman et al., 2012; Ladicky et al., 2014). Our proposed Diff-2-in-1 outperforms the baseline with more accurate surface normal estimations, indicating that our integrated diffusion-based models excel at handling discriminative tasks. The black regions in the ground truth visualizations are invalid regions.

uses RGB-class pairs to boost the performance. *Gaussian Noise Augmentation (GNA)* is a self-constructed baseline that generates additional data by denoising from Gaussian noise, then applies the self-improving strategy to utilize the generated data.

With the results shown in Table 1 and Figure 5, we observe: **(1)** When applying our Diff-2-in-1 on top of the state-of-the-art baselines, we achieve significantly better performance with notably fewer training data, demonstrating the great advantages of data efficiency from an integrated diffusion model. **(2)** Our Diff-2-in-1 has better performance than other augmentation methods like GNA and DA-Fusion, showcasing the usefulness of the multi-modal data generated by our pipeline, and the effectiveness of synthetic data utilization with our self-improving mechanism. **(3)** Our Diff-2-in-1 is a general design that can universally bring benefits to different discriminative backbones.

**Semantic segmentation.** We instantiate our Diff-2-in-1 on VPD (Zhao et al., 2023), a diffusion-based segmentation model. For self-improving, we synthesize one sample for each image in the training set. With the results shown in Table 2, we observe that the diffusion-based VPD can benefit from our paradigm by effectively performing self-improvement to leverage the generated samples.

**Multi-task evaluations.** We apply our Diff-2-in-1 on two state-of-the-art multi-task methods, In-vPT (Ye & Xu, 2022) and TaskPrompter (Ye & Xu, 2023). A total of 500 synthetic samples are generated for NYUD-MT following the surface normal evaluation. For PASCAL-Context, one sample is synthesized for each image in the training set with our Diff-2-in-1. The comparisons on NYUD-MT and PASCAL-Context are shown in Table 3 and Table 4, respectively. The results validate that our Diff-2-in-1 is a versatile design that can elevate the performance of a wide variety of vision tasks.

| Model | Source → Target | 11.25° (↑) | 22.5° (↑) | 30° (↑) | Mean (↓) | Median (↓) | RMSE (↓) |
|---|---|---|---|---|---|---|---|
| Bae et al. (2021) | ScanNet → NYUv2 | 59.0 | 77.5 | 83.7 | 16.0 | 8.4 | 24.7 |
| | NYUv2 → NYUv2 | 62.2 | 79.3 | 85.2 | 14.9 | 7.5 | 23.5 |
| Diff-2-in-1 on Bae et al. (2021) (Ours) | ScanNet → NYUv2 | **63.0** | **80.4** | **86.0** | **14.6** | **7.3** | **23.3** |

Table 6: Cross-domain evaluation on the surface normal estimation task of NYUv2 (Silberman et al., 2012; Ladicky et al., 2014). The performance of our method trained on ScanNet even outperforms the baseline Bae *et al.* trained on NYUv2, suggesting our generalizability to unseen datasets.

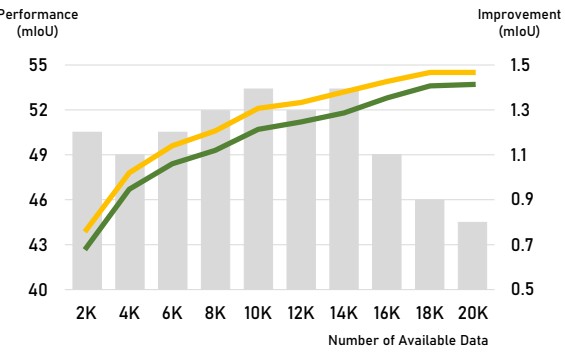

Figure 6: Ablation study on different data settings with our Diff-2-in-1. *Green line*: Performance of the baseline VPD. *Yellow line*: Performance with our Diff-2-in-1. *Gray bars*: Improvement in each data setting. Our Diff-2-in-1 consistently brings performance gain for all different data settings with more benefits in mid-range data settings.

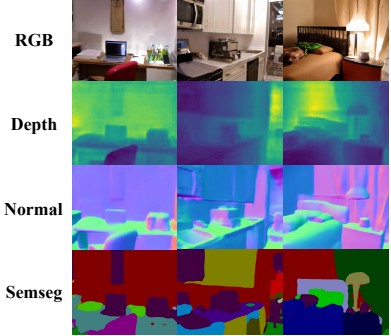

Figure 7: Multi-modal samples generated by our Diff-2-in-1 on NYUD-MT (Silberman et al., 2012). Our method generates high-quality and precise multi-modal data, thereby facilitating discriminative learning.

## 5.3 ABLATION STUDY

In this section, we offer a better understanding of the superiority of our Diff-2-in-1 by answering the three primary questions. More ablations are included in Section B in the appendix.

**How does timestep $T$ in data creation affect final performance?** As illustrated in Figure 4, the timestep $T$ balances the trade-off between the content variation and domain shift of the generated data. We ablate different timesteps $T \in \{300, 600, 800\}$ in the experiments on surface normal instantiated on Bae et al. (2021) and iDisc (Piccinelli et al., 2023). The results in Table 5 indicate that we achieve the best performance when $T = 600$, with a balance of data diversity and quality. Nevertheless, it is noteworthy that our performance is generally robust to different choices of $T$.

**How robust is Diff-2-in-1 for domain shift?** We perform the cross-domain evaluation to show that our Diff-2-in-1 has strong generalizability. We train both the baseline Bae et al. (2021) and our Diff-2-in-1 on the ScanNet (Dai et al., 2017) dataset for the surface normal estimation task, and evaluate the performance on the test set of NYUv2 (Silberman et al., 2012; Ladicky et al., 2014). Interestingly, with the results shown in Table 6, we find that the performance of our method trained on ScanNet even outperforms the baseline Bae *et al.* trained on NYUv2, suggesting the generalizability of our method to unseen datasets and its great potential in real practice.

**How Diff-2-in-1 is helpful in different data settings?** We ablate different settings when the number of available training samples for Diff-2-in-1 varies to investigate whether it is more helpful in data abundance or data shortage scenarios. We run this ablation for semantic segmentation on the ADE20K dataset: we randomly select 10% (2K) to 90% (18K) samples with 10% (2K) intervals in between, assuming that Diff-2-in-1 only gets access to partial data. In each setting, one additional sample for each image is generated using our data generation scheme.

With the results shown in Figure 6, we offer the following observations: (1) Diff-2-in-1 consistently boosts the performance under all settings, with improvements ranging from 0.8 to 1.4 in mIoU, indicating its effectiveness and robustness. (2) Diff-2-in-1 provides more benefits in data settings from 40% (8K) to 70% (14K). We analyze the reasons including that when the data are scarce, it is relatively hard to train a good model to provide high-quality multi-modal synthetic data for self-improvement. Meanwhile, when the data are already adequate, there is less demand for more diverse data. Under both scenarios, the benefit of our method is still noticeable yet less significant.

| Setting | 11.25° (↑) | 22.5° (↑) | 30° (↑) | Mean (↓) | Median (↓) | RMSE (↓) |
|---|---|---|---|---|---|---|
| GT Only | 56.6 | 76.8 | 83.0 | 17.2 | 9.3 | 26.6 |
| GT + Syn (Before Self-improving) | 57.5 | **77.1** | **83.3** | 17.1 | 9.1 | **26.5** |
| GT + Syn (After Self-improving) | **57.8** | **77.1** | **83.3** | **17.0** | **9.0** | **26.5** |

Table 7: Comparison between two data settings. *GT Only*: Use real samples to train Bae et al. (2021) until converges. *GT + Syn*: Further finetune the converged model with real and synthetic samples. Synthetic data further boost the performance of a converged model, demonstrating their realism.

| Backbone | Setting | 11.25° (↑) | 22.5° (↑) | 30° (↑) | Mean (↓) | Median (↓) | RMSE (↓) |
|---|---|---|---|---|---|---|---|
| Bae et al. (2021) | Synthetic | 67.4 | 83.4 | **88.2** | **13.2** | **6.5** | **22.0** |
| | Real | **67.5** | **83.5** | **88.2** | **13.2** | **6.5** | **22.0** |
| iDisc (Piccinelli et al., 2023) | Synthetic | **68.7** | **83.7** | **88.4** | **12.7** | **6.0** | 21.6 |
| | Real | **68.7** | **83.7** | **88.4** | 12.8 | **6.0** | **21.5** |

Table 8: Comparison between using generated samples and unlabeled real images in NYUv2 surface normal estimation. Comparable performance proves the premium quality of our generated data.

## 5.4 SYNTHETIC DATA EVALUATION

In addition to Figure 4, we visualize samples generated by our method on NYUD-MT (Silberman et al., 2012) in Figure 7. Diff-2-in-1 is able to generate high-quality RGB images and precise multi-modal annotations, further facilitating discriminative learning via our self-improvement. More qualitative visualizations can be found in Section C in the appendix. Below, we additionally examine the realism and usefulness of the generated data.

**Generated samples serving as data augmentation.** We select surface normal estimation as the target task and train an external discriminative model, Bae et al. (2021), under the following two settings: **(1)** only use the original 795 samples to train the model until convergence *(GT Only)*; and **(2)** finetune the converged model in *GT Only* using the mixture of original samples and generated samples from our Diff-2-in-1 before the self-improving stage *(GT + Syn)*. For (2), we generate 500 synthetic samples with $T = 600$ and naively merge them together with the original samples. We report two variants of setting (2) with generated samples before or after the self-improving stage in Table 7. We have the following observations: firstly, the synthetic samples are capable of boosting the performance of a converged model, indicating that the generated RGB and annotation maps are consistent. Moreover, the generated multi-modal data get refined during the self-improving stage, verifying the effectiveness of our method towards generation.

**Synthetic data V.S. real data.** In the surface normal task, we replace the 500 generated samples with 500 additional real captured images from NYUv2 raw video clips. The annotations of them are produced by our Diff-2-in-1 on the fly. Then, we use the same training strategy to train Diff-2-in-1. As shown in Table 8, using our generated data achieves comparable performance to using the real captured data, proving the premium quality of the synthetic data.

**Beyond multi-modal data evaluation.** In the evaluations above, we demonstrate the superior quality of our generated *multi-modal* data as a whole because we freeze the denoising U-Net in the majority of our experimental settings as discussed in Section A.3. As a result, the quality of the generated RGB images always remain the same. To further study the mutual benefits of the joint modeling of generative and discriminative learning within our Diff-2-in-1, we also explore the scenario when the denoising U-Net is unfrozen during training and how the performance of RGB generation will change accordingly. More details can be referred in Section D.3.

## 6 CONCLUSION

In this paper, we bridge generative and discriminative learning by proposing an integrated diffusion-based framework Diff-2-in-1. It enhances discriminative learning through the generative process by creating diverse while faithful data, and gets the discriminative and generative processes to interplay with each other using a self-improving learning mechanism. Extensive experiments demonstrate its superiority in various settings of discriminative tasks, and its ability to generate high-quality multi-modal data characterized by both realism and usefulness.

ACKNOWLEDGMENTS

This work was supported in part by NSF Grant 2106825, NIFA Award 2020-67021-32799, the Toyota Research Institute, the IBM-Illinois Discovery Accelerator Institute, the Amazon-Illinois Center on AI for Interactive Conversational Experiences, Snap Inc., and the Jump ARCHES endowment through the Health Care Engineering Systems Center at Illinois and the OSF Foundation. This work used computational resources, including the NCSA Delta and DeltaAI supercomputers through allocations CIS220014 and CIS230012 from the Advanced Cyberinfrastructure Coordination Ecosystem: Services & Support (ACCESS) program, as well as the TACC Frontera supercomputer and Amazon Web Services (AWS) through the National Artificial Intelligence Research Resource (NAIRR) Pilot.

CODE OF ETHICS

There is no obvious negative societal impact from our work. The potential negative impact is likely the same as other research on data generation with the risk of digital forgery.

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

In the appendix, we first include additional implementation details in Section A. Then, in Section B, we perform additional ablations on different implementation choices of text prompters, timesteps to perform discriminative learning with Diff-2-in-1, *etc.*, to provide more informative guidelines about how to apply our Diff-2-in-1 on discriminative tasks. Afterwards, we provide additional qualitative results in Section C, including comparisons of the performance on discriminative tasks and the multi-modal generation quality of our proposed Diff-2-in-1. Moreover, we include more experimental comparisons with diffusion-based methods (Ke et al., 2024) which repurpose text-to-image diffusion models for discriminative perception, more applications of our model on visual perception tasks beyond dense perception, and mutual benefits from our model when unfreezing the denoising network in Section D. Finally, we present dicussions of limitations and future work in Section E.

## A  IMPLEMENTATION DETAILS

### A.1  ARCHITECTURE DETAILS

**Feature extraction from diffusion models.** We first describe how we extract features for downstream dense prediction tasks from the pretrained stable diffusion model (Rombach et al., 2022) in our framework, which is generally applicable to all the model instantiations discussed below. We take the latent vector obtained from the VAE encoder in stable diffusion as input for the denoising network, followed by a one-step denoising to obtain the features. Since the denoising operation in stable diffusion is realized by a U-Net (Ronneberger et al., 2015) module, multi-scale features can be obtained through the one-step denoising process for a given image. As we use the publicly released stable diffusion pretrained weight `Stable Diffusion v1-5` which is finetuned on $512 \times 512$ resolution, the input images are also resized to $512 \times 512$ before being processed by our model. Therefore, the raw multi-scale features $\{f_i^{\mathrm{raw}}\}_{i=0}^3$ extracted from our model are in the spatial resolutions of $8 \times 8$, $16 \times 16$, $32 \times 32$, and $64 \times 64$. Following Li et al. (2023c), for each pair of features $f_{i-1}^{\mathrm{raw}}, f_i^{\mathrm{raw}}(1 \leq i \leq 3)$ with adjacent resolutions, we upsample the lower-resolution feature to the higher-resolution one, concatenating them, and processing with a convolutional layer:

$$f_i^{\mathrm{proc}} = \mathrm{Conv}(\mathrm{Up}(f_{i-1}^{\mathrm{raw}}), f_i^{\mathrm{raw}}). \tag{6}$$

Then, we get the processed multi-scale features $\{f_i^{\mathrm{proc}}\}_{i=1}^3$ which are further used for fitting into the specific network architectures when we build our Diff-2-in-1 on existing works.

**Surface normal estimation.** For both Bae et al. (2021) and iDisc (Piccinelli et al., 2023), the surface normal maps are decoded from multi-scale features extracted by their original encoder. When instantiating our Diff-2-in-1 upon them, we replace their original encoders with the integrated model described above. If the decoder requires a feature map with a spatial resolution unavailable in $\{f_i^{\mathrm{proc}}\}_{i=1}^3$, we use a similar strategy as Equation 6 to obtain the feature of a new spatial resolution. If the features required are of higher resolution than the existing features, then we increase the resolution range of the features by

$$f_{i+1}^{\mathrm{proc}} = \mathrm{Conv}(\mathrm{Up}(f_i^{\mathrm{proc}}), \mathrm{Deconv}(f_i^{\mathrm{proc}})), \tag{7}$$

where the upsampling and deconvolutional (Noh et al., 2015) layers increase the feature size by the same ratio. For obtaining lower resolution features, we simply replace the upsampling and deconvolutional layers in Equation 7 with downsampling and convolutional layers. The upsampling or downsampling factor in Equation 7 is set to 2. Moreover, we can iteratively perform Equation 7 multiple times if the required features are more than twice larger or smaller than the features $\{f_i^{\mathrm{proc}}\}_{i=1}^3$ from the pretrained stable diffusion model.

**Semantic segmentation.** As VPD (Zhao et al., 2023) also builds upon stable diffusion (Rombach et al., 2022), we directly apply the self-improving algorithm in our Diff-2-in-1 on VPD to boost its performance.

**Multi-task learning.** The decoder of InvPT (Ye & Xu, 2022) requires multi-scale features. Therefore, we use the same strategy as the surface normal estimation methods (Bae et al., 2021; Piccinelli et al., 2023) to provide the decoder with the required features. The decoder of TaskPrompter (Ye & Xu, 2023) only requires single-scale features. Therefore, we use Equation 7 to resize all the features

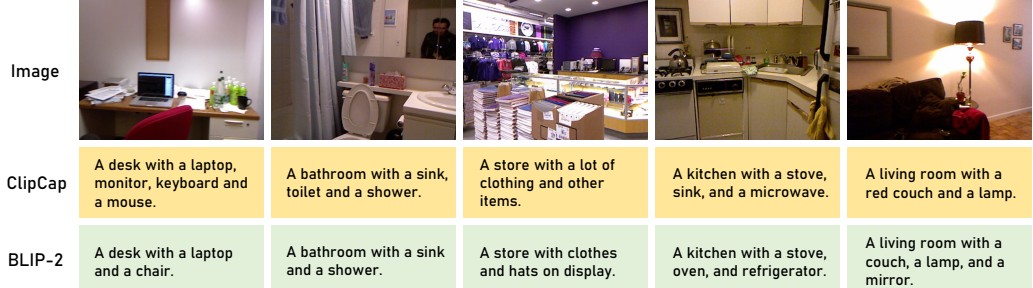

Figure A: Captions generated by ClipCap (Mokady et al., 2021) and BLIP-2 (Li et al., 2023b) on the NYUv2 (Silberman et al., 2012) dataset. The generated captions using these two off-the-shelf image captioning models not only have similar semantic meanings, but also share similar text formats.

in $\{f_i^{\text{proc}}\}_{i=1}^3$ to this specific scale. As a result, the multi-scale knowledge extracted from stable diffusion can be injected into the TaskPrompter framework. Additionally, both InvPT and TaskPrompter adopt pretrained ViT (Dosovitskiy et al., 2021) or Swin Transformer (Liu et al., 2021b) as their encoders. To better utilize the prior knowledge within the original encoders, we merge the knowledge from the two sources by adding the features from stable diffusion to their original encoders.

**Summary.** From the instantiations above, we have the following guidelines for converting existing methods to the integrated diffusion-based models in our Diff-2-in-1: **(1)** By default, we replace the encoders in the original models with the stable diffusion feature extractor; **(2)** If the features required by the original decoder is unavailable in the multi-scale features, we can use Equation 7 to expand the range of the multi-scale features; **(3)** If the original model design contains a pretrained encoder, we consider merging the knowledge of the stable diffusion model and the pretrained encoder.

## A.2 TEXT PROMPTS

Our Diff-2-in-1 uses the generative nature of diffusion models to create samples, which requires text prompts as conditions during the denoising process to generate high-quality samples. However, the text prompts are not always available in our target datasets. To solve this challenge, we use the off-the-shelf image captioning model BLIP-2 (Li et al., 2023b) to generate text descriptions for each image. The generated text descriptions serve as conditions when performing denoising to generate new data samples with our Diff-2-in-1. We further show in the ablation study in Section B that the choice of the image captioning model has little influence on the performance.

## A.3 ADDITIONAL TRAINING DETAILS

In the warm-up stage, we follow the same hyperparameters of the learning rate, optimizer, and training epochs of the original works that our Diff-2-in-1 builds on. In the self-improving stage, the exploitation parameter $\theta_E$ continues the same training scheme in the warm-up stage, while the creation parameter $\theta_C$ updates once when $\theta_E$ consumes 40 samples. Thus, the interval of the EMA update for $\theta_C$ depends on the batch size used in the self-improving stage. For the surface normal estimation and semantic segmentation tasks, we adopt a batch size of 4, so the EMA update happens every 10 iterations. For the multi-task frameworks, the batch size is 1, so we perform the EMA update every 40 iterations. The momentum hyperparameter $\alpha$ for the EMA update is set as 0.999 for multi-task learning on PASCAL-Context (Mottaghi et al., 2014), and 0.998 for the rest of the task settings. During the whole training process, we freeze the parameters in diffusion models by default, and only the parameters in the lightweight task head is tunable. The only exception in our experiments is when we build upon VPD for semantic segmentation, we follow the same setting as VPD to also allow the diffusion parameters to be trainable for fair comparison.

| Model | Caption | 11.25° (↑) | 22.5° (↑) | 30° (↑) | Mean (↓) | Median (↓) | RMSE (↓) |
|---|---|---|---|---|---|---|---|
| | None | 66.0 | 83.0 | 88.0 | 13.6 | 7.0 | **22.0** |
| Diff-2-in-1 on Bae et al. (2021) | ClipCap (Mokady et al., 2021) | 67.3 | **83.4** | **88.2** | **13.2** | **6.5** | **22.0** |
| | BLIP-2 (Li et al., 2023b) | **67.4** | **83.4** | **88.2** | **13.2** | **6.5** | **22.0** |
| | None | 67.2 | 83.4 | 88.1 | 13.0 | 6.6 | 21.7 |
| Diff-2-in-1 on iDisc (Piccinelli et al., 2023) | ClipCap (Mokady et al., 2021) | **68.7** | **83.7** | **88.4** | **12.7** | **6.0** | **21.6** |
| | BLIP-2 (Li et al., 2023b) | **68.7** | **83.7** | **88.4** | **12.7** | **6.0** | **21.6** |

Table A: Ablation study on using text prompts from different off-the-shelf image captioning models ClipCap (Mokady et al., 2021) and BLIP-2 (Li et al., 2023b) to generate samples with Diff-2-in-1. The evaluation is conducted on the surface normal estimation task on the NYUv2 (Silberman et al., 2012; Ladicky et al., 2014) dataset. Our Diff-2-in-1 is robust to different choices of image captioning models. Nevertheless, it is necessary to have an image captioning model to provide text prompts in the denoising process during data generation.

## B    ADDITIONAL ABLATION STUDY

**What text prompts to use for the integrated diffusion model?** As mentioned in Section A.2, we adopt BLIP-2 to generate text prompts for creating new samples based on the reference images. *What if the text prompters are less powerful?* We show that different choices of image captioning models have a marginal influence on the performance of our Diff-2-in-1. We first show the captions generated by BLIP-2 and another relatively weaker model ClipCap (Mokady et al., 2021) in Figure A. The captions generated by these two off-the-shelf models have similar semantic meanings, as well as sharing similar formats of *"A [Place] with [Object 1], [Object 2], ..., [Object N-1], and [Object N]."* We further evaluate the performance of using the text prompts from ClipCap and BLIP-2 to generate synthetic samples for the self-improving learning system in Diff-2-in-1. The results are shown in Table A. We can observe that once again there is no large difference between the two variants and both of them greatly outperform the baseline, demonstrating that our Diff-2-in-1 is robust to different text prompters used during the denoising process for data generation. Nonetheless, it does not indicate that the image captioning model is dispensable. If we completely get rid of the image captioning model and do not use text as the condition during denoising (*None* for *Caption* in Table A), we could observe an evident drop in the performance on discriminative tasks.

**Should we finetune the diffusion backbone?** As shown in Figure 3, if the generation process of our integrated diffusion model starts from Gaussian noise, the generated samples will have an evident domain shift from the original distribution. Therefore, we adopt the halfway diffusion-denoising mechanism to synthesize in-distribution data. Another potential solution to overcome the domain shift issue is to finetune the stable diffusion backbone. We test this setting with two finetuning strategies for a comprehensive ablation: (1) directly finetune all the parameters of the denoising U-Net *(Direct Finetuning)*; (2) adopt parameter-efficient finetuning strategy Low-Rank Adaptation (LoRA) (Hu et al., 2022) on the denoising modules of stable diffusion *(LoRA Finetuning)*. We conduct the experiments on the surface normal task on the NYUv2 dataset with Bae et al. (2021) as the task head. The results are shown in Table B. The inferior performance of using the finetuned stable diffusion indicates that the diffusion-denoising data generation scheme and the self-improving learning system in our Diff-2-in-1 are essential. One factor for the unsatisfactory performance of using finetuning is that the finetuning process incurs a loss in the generalization capability, especially during finetuning with limited data (*e.g.*, 795 samples on NYUv2), making the features extracted from the stable diffusion model less informative for decoding visual task predictions. In comparison, our proposed diffusion-denoising data generation scheme injects external knowledge from the pretrained stable diffusion model to the samples in the training data, without risks of knowledge forgetting with respect to its discriminative ability.

**What timestep $T$ to choose for discriminative feature extraction?** In our current experiments, we follow existing works ODISE (Xu et al., 2023a) and VPD (Zhao et al., 2023) to adopt $T = 0$ as the timestep for feature extraction from the pretrained stable diffusion model. We ablate different timesteps $T$ for extracting features from stable diffusion in Table C. The performance is generally satisfactory with relatively small timesteps $T$, which add little noise to the clean latents before extracting features from denoising U-Net. We do not attentively optimize for the best $T$ and it is likely that a better $T$ may exist in other settings which can further improve the performance of our Diff-2-in-1. We leave the exploration of optimal $T$ for different tasks as future work.

| Setting | 11.25° (↑) | 22.5° (↑) | 30° (↑) | Mean (↓) | Median (↓) | RMSE (↓) |
|---|---|---|---|---|---|---|
| Direct Finetuning | 58.0 | 76.5 | 82.4 | 16.9 | 8.7 | 26.5 |
| LoRA Finetuning | 64.8 | 82.0 | 87.4 | 14.1 | 7.3 | 22.8 |
| Diff-2-in-1 (Ours) | **67.4** | **83.4** | **88.2** | **13.2** | **6.5** | **22.0** |

Table B: Ablation study on strategies to finetune the diffusion backbone. *Direct Finetuning*: Directly finetune the denoising U-Net. *LoRA Finetuning*: Adopt LoRA (Hu et al., 2022) to finetune the U-Net. Their unsatisfactory results indicate that the features extracted from the finetuned network are less informative and have worse generalizability. The information loss introduced by finetuning is inevitable even if using the parameter-efficient finetuning technique LoRA to mitigate forgetting. In contrast, our diffusion-denoising strategy injects external knowledge from the pretrained stable diffusion to the samples, without risks of forgetting the discriminative ability of diffusion models.

| $T$ | 11.25° (↑) | 22.5° (↑) | 30° (↑) | Mean (↓) | Median (↓) | RMSE (↓) |
|---|---|---|---|---|---|---|
| 0 | 67.4 | **83.4** | **88.2** | **13.2** | **6.5** | **22.0** |
| 50 | **67.5** | 83.3 | 88.1 | **13.2** | **6.5** | **22.0** |
| 100 | 66.9 | 82.6 | 87.5 | 13.5 | **6.5** | 22.4 |
| 150 | 65.5 | 81.6 | 86.7 | 14.0 | 6.8 | 23.0 |

Table C: Ablation study on extracting features from the pretrained stable diffusion model with different timesteps $T$ on NYUv2 surface normal evaluation. Our Diff-2-in-1 achieves better performance with smaller $T$ in this task setting.

**How to choose hyperparameters for the EMA update?** We ablate the choice of $\alpha \in [0.99, 0.999]$ for the EMA update according to guidelines in Liu et al. (2021a). The results with Bae et al. (2021) on the NYUv2 (Silberman et al., 2012; Ladicky et al., 2014) surface normal task are shown in Table D where $\alpha = 0.998$ achieves the best performance. Nevertheless, the performance of our Diff-2-in-1 is robust to different choices of $\alpha$ within a broad range.

**How important is the self-improving mechanism in Diff-2-in-1?** The self-improving learning mechanism is a key design in our framework. We ablate the usage of the self-improving stage in the surface normal experiment instantiated on Bae et al. (2021) in Table E, where *w/o self-improving* indicates that we discard our self-improving strategy, and instead mix the original and the generated samples to finetune the model after the warm-up stage. We observe that the self-improving mechanism indeed further boosts the performance of the model by fostering an effective interaction between the discriminative and generative components in our framework.

## C  MORE VISUALIZATIONS

We provide more qualitative results from the following two aspects: **(1)** performance comparison with state-of-the-art methods on discriminative tasks and **(2)** multi-modal data generation quality of the synthetic samples from our Diff-2-in-1.

### C.1  COMPARISONS ON DISCRIMINATIVE TASKS

The qualitative comparisons of our Diff-2-in-1 and the baselines are shown in Figures B, C (surface normal prediction) and D (multi-task). Our Diff-2-in-1 outperforms the baselines, demonstrating the competence of our integrated diffusion-based model in the discriminative perspective.

### C.2  DATA GENERATION QUALITY

We display the synthetic multi-modal data from our Diff-2-in-1 data creation framework in Figures E, F (RGB-normal pairs) and G, H (RGB and multiple annotations) to show that Diff-2-in-1 has powerful generation ability that is capable of generating high-quality and consistent samples.

| $\alpha$ | 11.25° (↑) | 22.5° (↑) | 30° (↑) | Mean (↓) | Median (↓) | RMSE (↓) |
|---|---|---|---|---|---|---|
| *N/A* (Baseline) | 62.2 | 79.3 | 85.2 | 14.9 | 7.5 | 23.5 |
| 0.99 | 67.1 | 83.2 | 88.1 | 13.4 | 6.6 | 22.1 |
| 0.993 | 67.3 | **83.4** | **88.2** | 13.3 | 6.6 | **22.0** |
| 0.996 | 67.3 | **83.4** | **88.2** | 13.3 | 6.6 | **22.0** |
| 0.998 | **67.4** | **83.4** | **88.2** | **13.2** | **6.5** | **22.0** |
| 0.999 | 67.1 | 83.3 | 88.1 | 13.3 | 6.7 | 22.1 |

Table D: Ablation study on different $\alpha$ for the EMA update within Diff-2-in-1. $\alpha = 0.998$ reaches the best performance in this setting of surface normal prediction with Bae et al. (2021) on NYUv2. Nonetheless, our Diff-2-in-1 is robust to different $\alpha$ within a broad range.

| Setting | 11.25° (↑) | 22.5° (↑) | 30° (↑) | Mean (↓) | Median (↓) | RMSE (↓) |
|---|---|---|---|---|---|---|
| w/o Self-improving | 65.2 | 82.4 | 87.5 | 14.0 | 7.2 | 22.7 |
| Diff-2-in-1 (Ours) | **67.4** | **83.4** | **88.2** | **13.2** | **6.5** | **22.0** |

Table E: Ablation study of the self-improving mechanism. The self-improving strategy further boosts the performance of the model by fostering an effective interaction between the discriminative and generative components in our framework.

# D    ADDITIONAL EXPERIMENTAL COMPARISONS

## D.1    COMPARISON WITH MARIGOLD

As discussed in the related work, another line of work repurposes diffusion models for dense prediction by finetuning the denoising network. We include additional comparisons with these diffusion-based dense perception methods, empirically demonstrating that our framework is more flexible for such tasks. We choose Marigold (Ke et al., 2024) as an example, due to its most relevance and most complete codebase. We make comparisons with three variants of Marigold on the NYUv2 surface normal estimation benchmarks. **Marigold (Pretrain)** is the released checkpoint that is trained on a mixed large dataset excluding NYUv2. **Marigold (SD)** is obtained when we adopt the same training setting as our framework, using the 795 training samples to train Marigold from the Stable Diffusion checkpoint until convergence. **Marigold (Finetune)** is obtained by further finetuning the released checkpoint with 795 training samples from NYUv2.

The comparison is shown in Table F. Notably, all three variants lagged behind our model, indicating the effectiveness of our model design with the self-improving learning mechanism. Moreover, we observe that Marigold gets inferior performance when adapted to a specific domain with a limited amount of training data (795 samples). While finetuning from a well-trained model can help mitigate this issue, it still does not work as well as our proposed method. The reason is that tuning these diffusion-based perception models like Marigold, which require finetuning the denoising U-Net, is computationally expensive. In comparison, our approach only requires training a lightweight task head, which makes our framework more flexible and easier to train or fine-tune for new domains.

In addition, we report the comparison of the computational cost of our method and Marigold with a batch of images of shape (2, 512, 512, 3) in Table G. Our framework is a more efficient and effective solution compared with Marigold.

## D.2    APPLICATION ON PERCEPTION TASKS BEYOND DENSE PERCEPTION

Despite the focus of our work being dense perception tasks, which are a series of primary and important tasks in computer vision, our framework is a general-purpose design that can be easily applied to other visual perception tasks beyond dense prediction. We perform the following experiment on the multi-task setting in a subset of Tiny-Taskonomy (Zamir et al., 2018) including the scene categorization task which is beyond dense perception. The results in Table H validate that our framework can also provide improvement on other types of perception tasks beyond dense pixel prediction.

## D.3    MUTUAL BENEFITS WHEN UNFREEZING THE DENOISING NETWORK

As mentioned in Section A.3, we freeze the denoising U-Net and only train the task head for dense perception in the majority of our experimental settings. Therefore, our evaluations focus on the

| Model | 11.25° (↑) | 22.5° (↑) | 30° (↑) | Mean (↓) | Median (↓) | RMSE (↓) |
|---|---|---|---|---|---|---|
| Marigold (Pretrain) | 50.5 | 73.0 | 79.3 | 20.9 | 11.1 | 26.2 |
| Marigold (SD) | 48.7 | 76.8 | 84.0 | 18.1 | 11.5 | 25.8 |
| Marigold (Finetune) | 64.0 | 82.4 | 87.8 | 14.2 | 7.7 | 22.3 |
| Diff-2-in-1 on Bae et al. (2021) (Ours) | **67.4** | **83.4** | **88.2** | **13.2** | **6.5** | **22.0** |

Table F: Comparison with diffusion-based visual perception method Marigold (Ke et al., 2024) on the NYUv2 surface normal benchmark. Our framework outperforms all three variants of Marigold, indicating that our framework is a superior choice with limited data for finetuning.

| Metrics | Training Time (s/iteration) (↓) | Model Size (M) (↓) | GPU Memory (GB) (↓) |
|---|---|---|---|
| Marigold | 1.08 | 860 | 30 |
| Diff-2-in-1 on Bae et al. (2021) (Ours) | **0.28** | **96** | **10** |

Table G: Comparison of computational costs between our method and Marigold. The numbers are reported by running a batch of images of shape (2, 512, 512, 3). Our framework is a more efficient and effective solution compared with Marigold.

performance on the dense perception tasks and the quality of the generated multi-modal data, which are the main targets of our paper. In this section, we unfreeze the denoising U-Net to study whether our joint modeling of discriminative and generative learning can provide benefits for both *RGB generation* and dense perception. More specifically, when unfreezing the denoising U-Net in the diffusion model, we add an additional loss term on noise prediction for RGB generation alongside our original discriminative loss. We do not consciously tune the weight of the additional loss term, and simply adopt a default weight of 1.0.

We conduct the experiment for the surface normal task on Bae et al. (2021). For RGB generation, we evaulate on the set of synthetic images generated for the self-improving stage with the metric of FID (Heusel et al., 2017). After unfreezing the denoising U-Net and jointly modeling the discriminative and generative objectives with our integrated Diff-2-in-1 model, the FID improves from 48.53 to 40.40. Meanwhile, for the surface normal prediction task, the comparison on the performance is shown in Table I. Unfreezing the denoising U-Net during training brings additional burden for computation, but it further boosts the performance of the discriminative task by an evident margin. It demonstrates the mutual benefit from the joint modeling of generative and discriminative learning within our Diff-2-in-1 model.

# E DISCUSSIONS AND FUTURE WORK

**Limitation.** One major limitation of this work is that adopting diffusion models for data generation is relatively time-consuming as diffusion models typically need multi-step denoising to produce samples. To alleviate this shortcoming, current advancement on accelerating the inference process of diffusion models (Zheng et al., 2023a; Lu et al., 2022; Yin et al., 2024; Liu et al., 2024b) can be adopted to speed up the data generation process.

**Future work.** Looking ahead, the potential applications of this integrated diffusion model are vast. Future research directions include extending this methodology to other types of tasks, such as 3D detection, and refining and optimizing the Diff-2-in-1 framework such as a more efficient data creation scheme and knowledge transfer to a new domain.

| Model | Categorization Top-1 Acc. (↑) | Semseg mIoU (↑) | Depth RMSE (↓) | Normal mErr (↓) |
|---|---|---|---|---|
| TaskPrompter | 38.80 | 15.63 | 0.8350 | 28.87 |
| Diff-2-in-1 on TaskPrompter (Ours) | **39.67** | **16.61** | **0.8289** | **28.35** |

Table H: Comparison on Tiny-Taskonomy (Zamir et al., 2018). Our Diff-2-in-1 can also provide improvement on other types of perception tasks beyond dense pixel prediction.

| Settings | 11.25° (↑) | 22.5° (↑) | 30° (↑) | Mean (↓) | Median (↓) | RMSE (↓) |
|---|---|---|---|---|---|---|
| Frozen | 67.4 | 83.4 | 88.2 | 13.2 | 6.5 | 22.0 |
| Unfrozen | **68.6** | **84.4** | **89.1** | **12.8** | **6.4** | **21.6** |

Table I: Comparison on the surface normal task with Bae et al. (2021). *Unfrozen* indicates the setting that the denoising U-Net is tunable during training, while *Frozen* is our original setting that freezes the denoising U-Net for more efficient training. The improvement from *Unfrozen* to *Frozen* demonstrates that our Diff-2-in-1 can further benefit from the joint modeling of discriminative and generative tasks when the computational budget is sufficient to support unfreezing the denoising network in diffusion models.

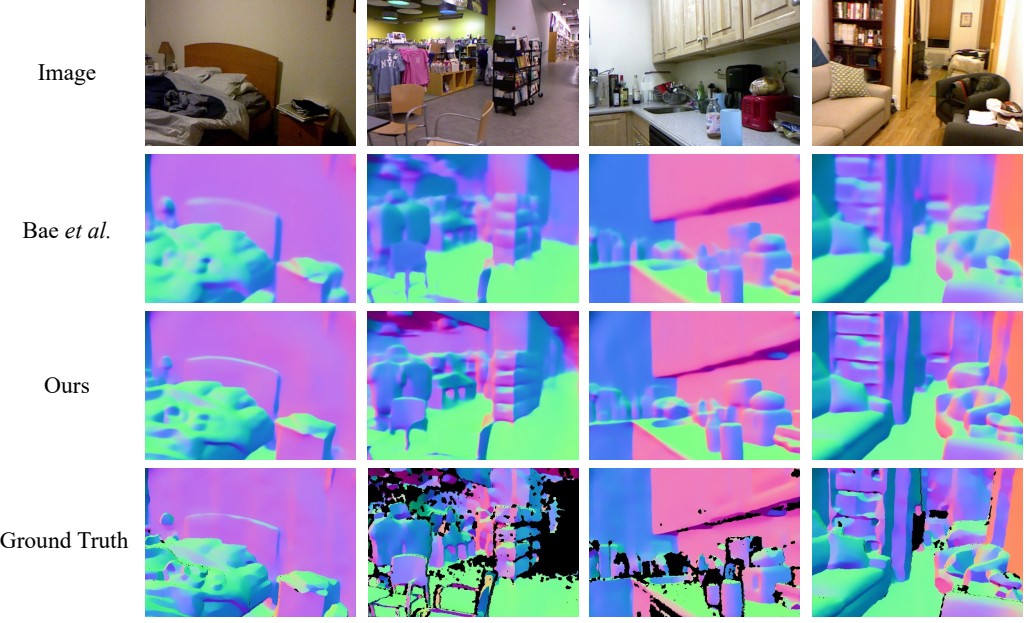

Figure B: Qualitative results on the surface normal prediction task of NYUv2 (Silberman et al., 2012; Ladicky et al., 2014). Our proposed Diff-2-in-1 outperforms the baseline with more accurate surface normal estimations, indicating that our integrated diffusion-based models excel at handling discriminative tasks. The black regions in the ground truth visualizations are invalid regions.

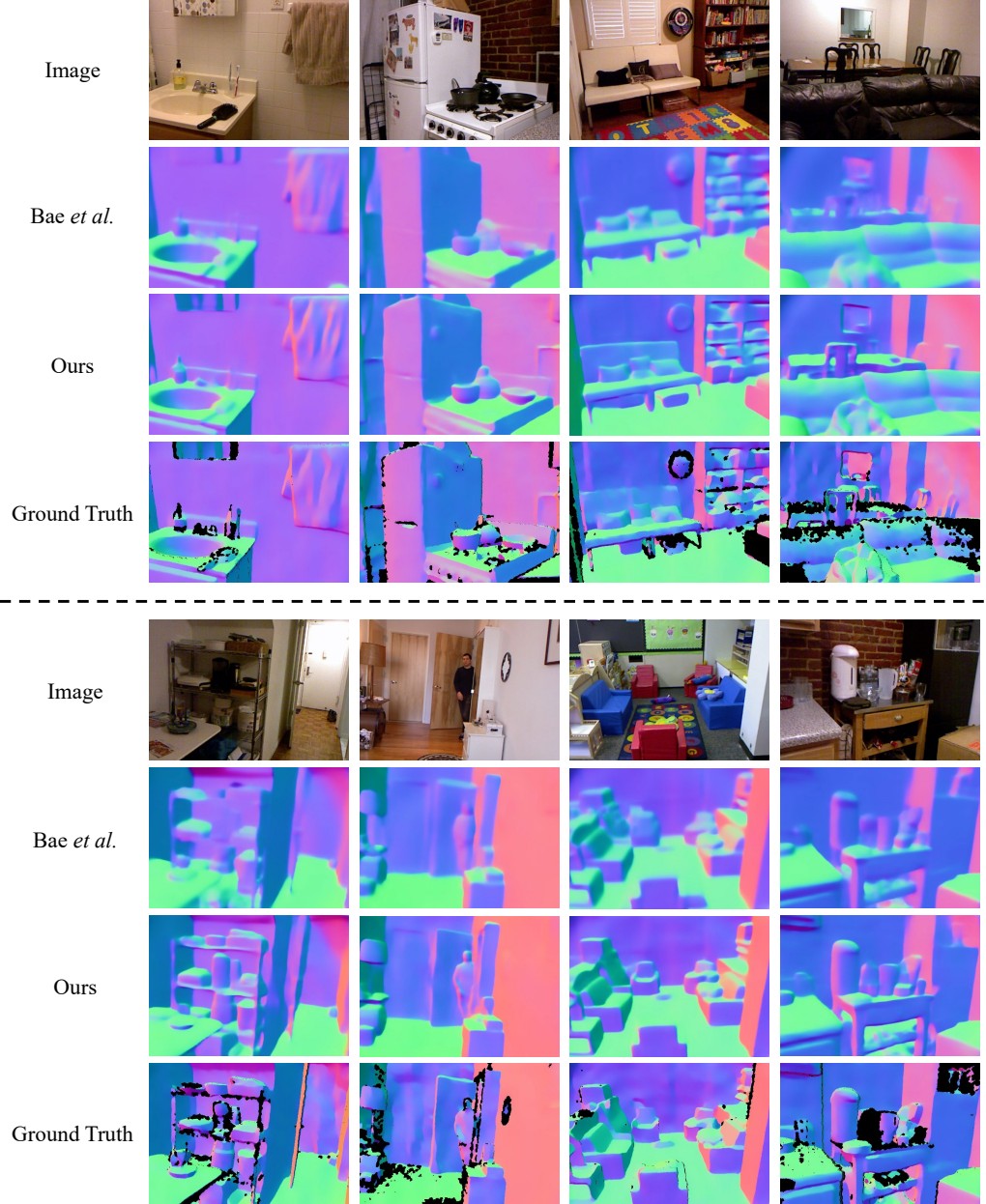

Figure C: Qualitative results on the surface normal task of NYUv2 (Silberman et al., 2012; Ladicky et al., 2014). Our proposed Diff-2-in-1 outperforms the baseline with more accurate surface normal estimations, indicating that our integrated diffusion-based models excel at handling discriminative tasks. The black regions in the ground truth visualizations are invalid regions.

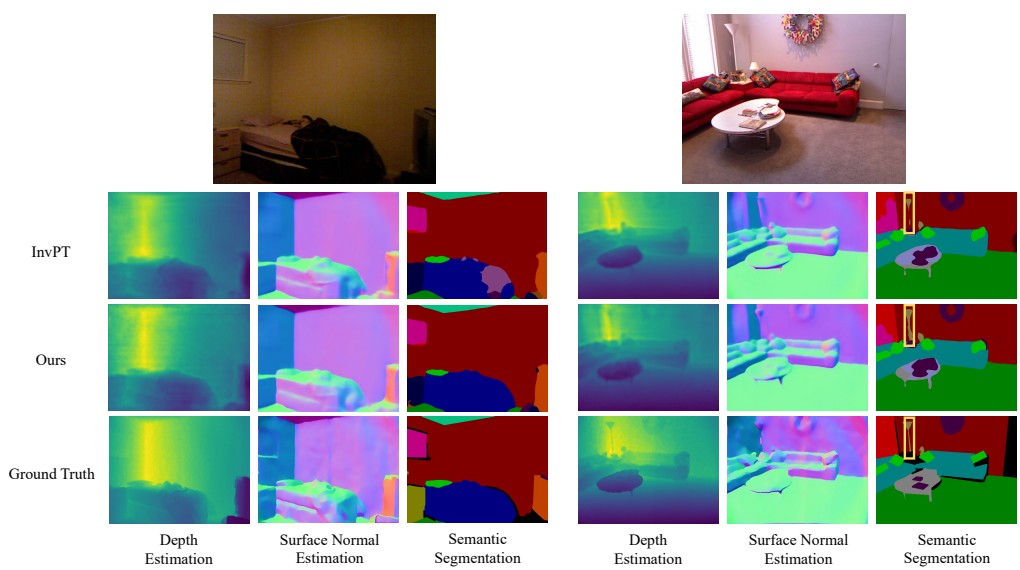

(a) Comparison on the NYUD-MT dataset

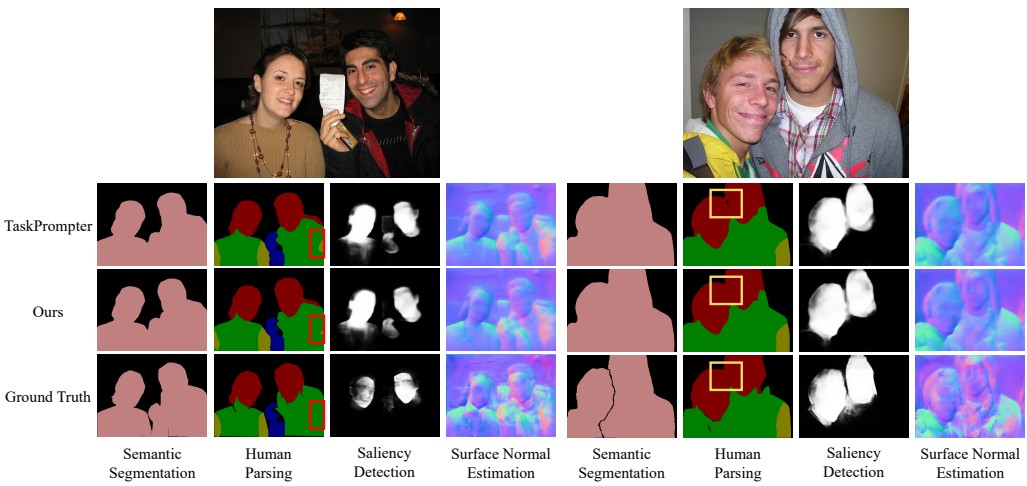

(b) Comparison on the PASCAL-Context dataset

Figure D: Qualitative results on the multi-task datasets NYUD-MT (Silberman et al., 2012) and PASCAL-Context (Mottaghi et al., 2014). Diff-2-in-1 has superior performance compared to the baselines, demonstrating the effectiveness of our integrated diffusion-based model design. Zoom in for the regions with bounding boxes to better see the comparison.

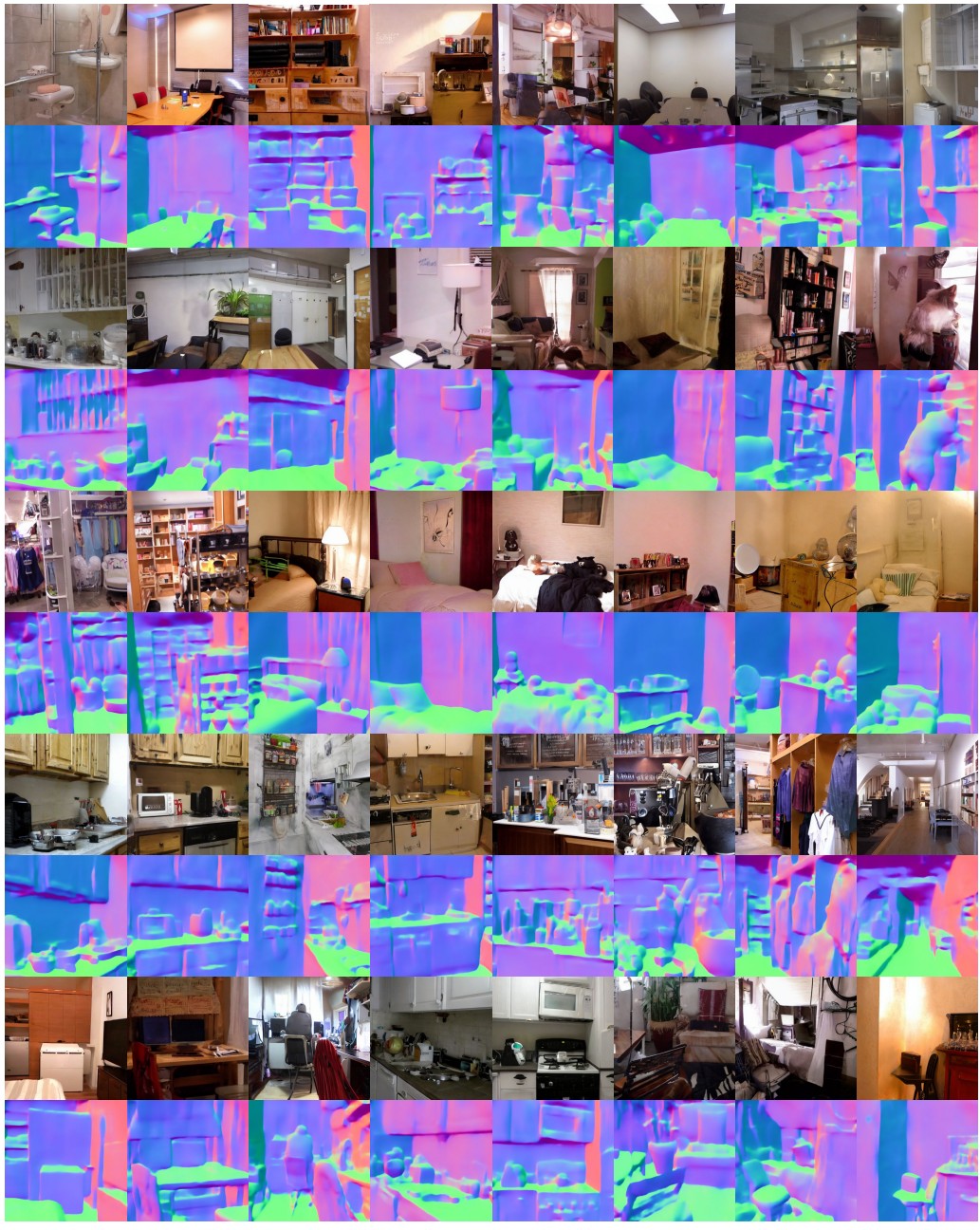

Figure E: Synthetic samples from our method after the Diff-2-in-1 framework is trained on the surface normal task of NYUv2 (Silberman et al., 2012; Ladicky et al., 2014). The odd rows are the generated RGB images while the even rows are the generated surface normal maps. The model is capable of generating diverse and high-fidelity images with the corresponding surface normal maps matching the generated RGB images.

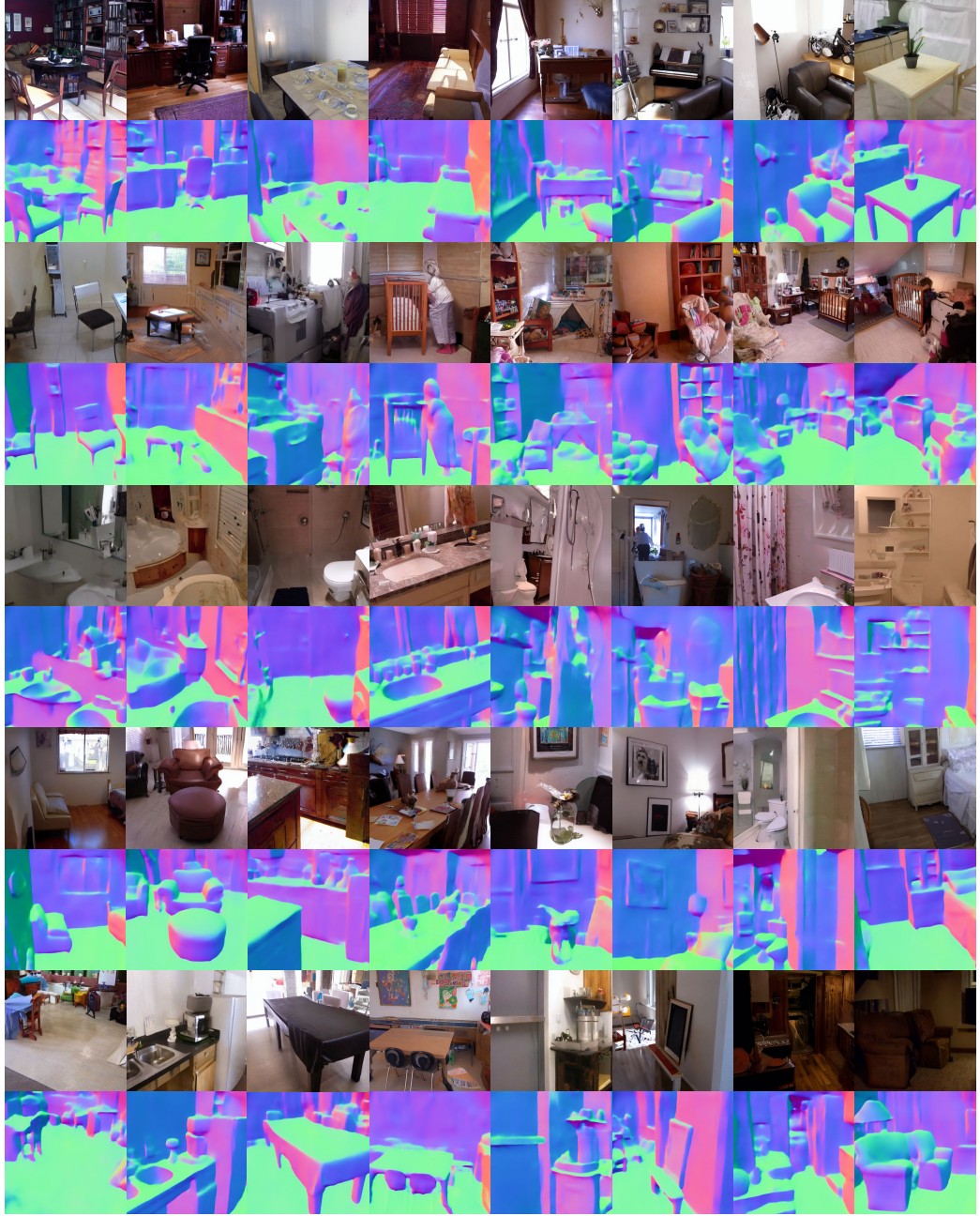

Figure F: Synthetic samples from our method after the Diff-2-in-1 framework is trained on the surface normal task of NYUv2 (Silberman et al., 2012; Ladicky et al., 2014). The odd rows are the generated RGB images while the even rows are the generated surface normal maps. The model is capable of generating diverse and high-fidelity images with the corresponding surface normal maps matching the generated RGB images.

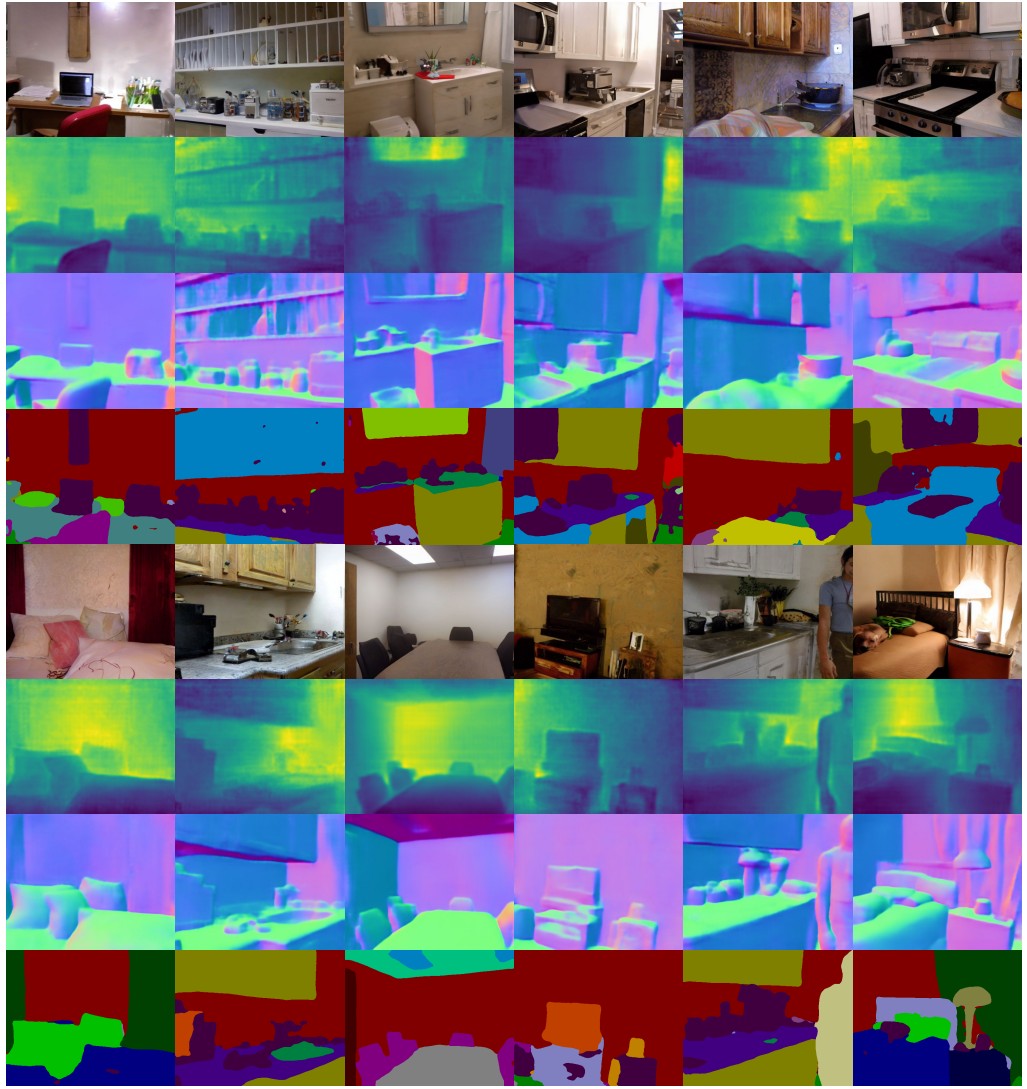

Figure G: Synthetic samples from our method after the Diff-2-in-1 framework is trained on the multi-task setting of NYUD-MT (Silberman et al., 2012). Each batch of samples contains four rows: RGB, depth map, surface normal map, and semantic labels *(from top to bottom)*. The generated samples are of high quality with their multi-task annotations.

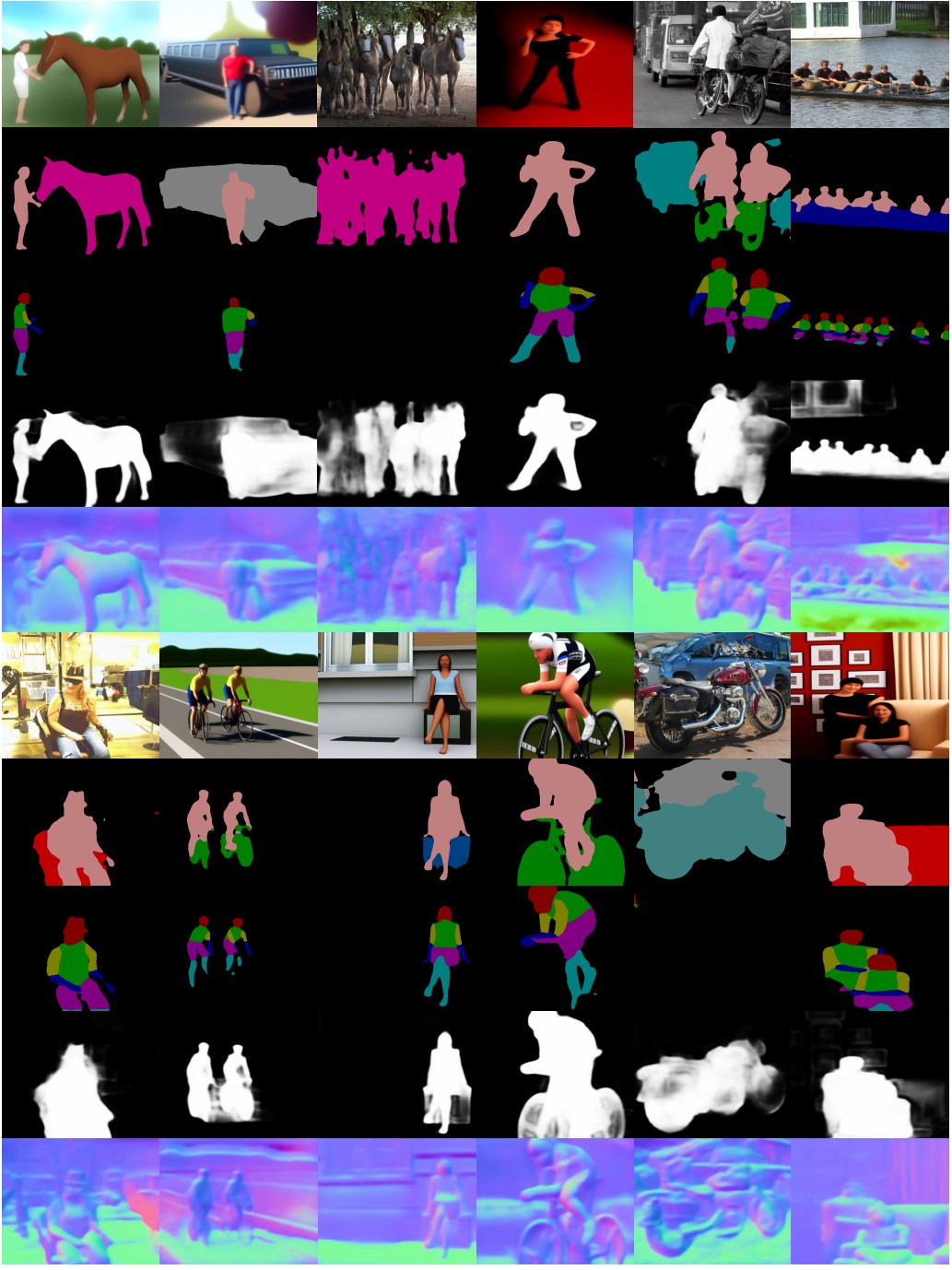

Figure H: Synthetic samples from our method after the Diff-2-in-1 framework is trained on the multi-task setting of PASCAL-Context (Mottaghi et al., 2014). Each batch of samples contains five rows: RGB, semantic labels, human parsing labels, saliency map, and surface normal map *(from top to bottom)*. If the human parsing labels are all black, it means that there is no human in the generated image. The generated samples are of high quality with their multi-task annotations.

