# OpenReview forum: "Diff-2-in-1: Bridging Generation and Dense Perception with Diffusion Models"
_ICLR.cc/2025/Conference — ICLR 2025 Poster_

### Official Review · Reviewer_8JiW · 2024-10-29

**Soundness:** 3
**Presentation:** 1
**Contribution:** 3
**Rating:** 6
**Confidence:** 3

**Summary:**

The paper proposes a new diffusion-based framework called Diff-2-in-1, which integrates both multi-modal data generation and dense visual perception. Unlike most prior work where diffusion models are used in isolated manners (either for generating data or for perception tasks), the authors introduce a unified approach that leverages the denoising process of diffusion models to simultaneously perform both tasks.

**Strengths:**

1. The idea of merging both data generation and visual perception into one unified framework is interesting and novel.
2. The method is versatile enough to handle multiple tasks such as segmentation, depth estimation and surface normals estimation. demonstrating wide applicability in many downstream tasks.

**Weaknesses:**

1. Both the figure (Figure 2) and writing for the section of self-improving learning are bad and seem to not be well-motivated. The performance gain of self-improving learning is not good either as the scores for most metrics remain unchanged (in Table 7)).

2. There is no clear breakdown of which components of the self-improving mechanism or the diffusion-denoising process contribute most to the observed improvements. An ablation study would help pinpoint the critical aspects of the framework. For example, performing an additional experiment without using the diffusion-denoising process.

3. For depth and surface normal estimation, the authors do not compare their method against Marigold [1] and StableNormal [2], which are very powerful models for these tasks

Reference:

[1] Ke et. al. Repurposing Diffusion-Based Image Generators for Monocular Depth Estimation. 2024

[2] Ye et. al. StableNormal: Reducing Diffusion Variance for Stable and Sharp Normal. 2024

**Questions:**

I have no further questions.

---

> ### Author Response · Authors · 2024-11-25
> **Response to Reviewer 8JiW [1/2]**
>
> We sincerely appreciate the detailed feedback you provide for our work! Below, we provide the following clarifications to address your concerns:
>
> ---
>
> **W1-1: Writing improvement on Figure 2 and Section 4.3**
>
> Thanks for your comment! The original Figure 2 contained some inconsistent notation in the color illustration as mentioned by Reviewer Got7's [W3], which may be the reason for the cause of confusion. We have addressed this issue by providing a corrected version of Figure 2 in the revised manuscript. We have also made revisions on Lines 264-265, 269, 311-314, 318-320 to clarify and better motivate the self-improving component. Basically, the underlying motivation of our proposed self-improving learning mechanism is that while synthetic multi-modal data typically demonstrates high visual fidelity, its direct utility for discriminative learning remains uncertain, so we need to have a self-improving stage to efficiently utilize the synthetic multi-modal data for discriminative perception tasks. We are glad to further revise the corresponding figures and texts if you have remaining concerns about them.
>
> ---
>
> **W1-2: Self-improving learning performance in Table 7**
>
> Thank you for your comment! First, we would like to kindly clarify that Table 7 is ***not*** intended to ablate the self-improving technique, but rather to assess the quality of the generated multi-modal data. More specifically,
>
> - In Table 7, we quantitatively evaluate the quality of the generated multi-modal data, by training an external discriminative model (Bae et al.) using a mixture of the original samples and generated samples. This evaluation serves as an “offline” assessment of the multi-modal data quality, and importantly, ***no*** self-improving learning is involved in the training of the external model (Bae et al.).
>
> - The results in Table 7 show that both "GT + Syn (Before Self-improving)" and "GT + Syn (After Self-improving)" outperform the basic version of "GT Only", demonstrating the ***high-quality and usefulness of the multi-modal data*** generated from our framework.
>
> - However, it is important to note that the comparison between "before/after self-improving" in Table 7 should not be interpreted as a rigorous ablation of the self-improving technique. This is because the setting in Table 7 does not leverage the full potential of the self-improving process --- in Table 7's setting, we only use the synthetic multi-modal data to train an external model (Bae et al.) without incorporating the closed-loop interaction between generation and discrimination.
>
>
> Second, we provide an ablation to directly ablate the effect of self-improving learning by directly comparing the variants of our model with and without the self-improving mechanism using Bae et al. on the NYUv2 surface normal benchmark, as below:
>
>
>
>
>
> |Settings|11.25$^\circ(\uparrow)$|22.5$^\circ(\uparrow)$|30$^\circ(\uparrow)$|Mean ($\downarrow$)|Median ($\downarrow$)|RMSE ($\downarrow$)|
> |:---|:---:|:----:|:----:|:----:|:---:|:---:
> |Full Model | **67.4** | **83.4** | **88.2** | **13.2** | **6.5** | **22.0** |
> |w/o self-improving | 65.2 | 82.4 | 87.5 | 14.0 | 7.2 | 22.7 |
>
> We can observe that the self-improving mechanism indeed further boosts the performance of the model, and the improvement is evident.
>
> ---
>
> **W2: Ablation study on the critical aspects of the framework**
>
> Thanks for the comment! Following the suggestions, we conducted the ablation study on whether or not to include self-improving learning, as shown in response to the previous concern W1.
>
> Moreover, we would like to kindly ask for clarification on the suggested ablation of "a variant without using diffusion-denoising process". Based on our understanding, it is hard to conduct such an ablation. The reason is that the unique denoising-diffusion process is integral to our  unified diffusion model, which enables both multi-modal data generation and dense pixel prediction. This dual capability further serves as the foundation for the self-improving learning mechanism --- a wise way to better utilize the generated multi-modal data to boost discriminative learning. Therefore, it is not feasible to remove the diffusion-denoising process from our current framework.
>
> We are glad to provide more ablations for a better understanding of our framework if the reviewer could further clarify or suggest other experiments.

---

> ### Author Response · Authors · 2024-11-25
> **Response to Reviewer 8JiW [2/2]**
>
> **W3: Comparison with Marigold [1] and StableNormal [2]**
>
> First, we would like to clarify that while our work, Marigold [1], and StableNormal [2] are all relevant to the surface normal estimation task, they pursue different exploration directions and aim to achieve distinct objectives. Marigold [1] and StableNormal [2] repurpose diffusion models for dense prediction by finetuning the denoising network, which results in the loss of the model's RGB generation capability. In contrast, our framework integrates both generation and discrimination, preserving the ability for RGB generation while further extending it to multi-modal generation.
>
> Second, we provide a comparison with methods that repurpose diffusion models for discriminative tasks, empirically demonstrating that our framework is more flexible for such tasks. Below, we mainly focus on a comprehensive comparison with Marigold [1], as StableNormal has not released the official training/finetuning code.
>
> - We compared three variants of Marigold on NYUv2 surface normal estimation in total. Variant 1, denoted as **Marigold (pretrain)**, is their released checkpoint that is trained on a mixed large dataset excluding NYUv2. For variant 2, we adopt the same setting as ours, using the 795 training samples to train Marigold from the Stable Diffusion checkpoint until convergence, denoted as **Marigold (SD)**. For variant 3, we further finetune their released checkpoint with 795 training samples from NYUv2, denoted as **Marigold (finetune)**. The results of the three variants, together with ours are reported below, as well as in Table F in the revised manuscript.
>
> |Settings|11.25$^\circ(\uparrow)$|22.5$^\circ(\uparrow)$|30$^\circ(\uparrow)$|Mean ($\downarrow$)|Median ($\downarrow$)|RMSE ($\downarrow$)|
> |:---|:---:|:----:|:----:|:----:|:---:|:---:
> |Marigold (pretrain) | 50.5 | 73.0 | 79.3 | 20.9 | 11.1 | 26.2|
> |Marigold (SD) | 48.8 | 76.8 | 84.0 | 18.1 | 11.5 | 25.8|
> |Marigold (finetune) | 64.0 | 82.4 | 87.8 | 14.2 | 7.7 | 22.3|
> |Diff2-in-1 on Bae et al. (Ours) | **67.4** | **83.4** | **88.2** | **13.2** | **6.5** | **22.0** |
>
>
> - All three variants lagged behind our model, indicating the effectiveness of our model design with the self-improving learning mechanism. Moreover, we observe that Marigold gets inferior performance when adapted to a specific domain with a limited amount of training data (795 samples). While finetuning from a well-trained model can help mitigate this issue, it still does not work as well as our proposed method. The reason is that tuning these diffusion-based perception models like Marigold, which require finetuning the denoising U-Net, is computationally expensive. In comparison, our approach only requires training a lightweight task head, which makes our framework more flexible and easier to train or fine-tune for new domains.
>
> - In addition, the comparison of the computational cost of our method and Marigold with a batch of images of shape (2, 512, 512, 3) is reported below:
>
> |Metrics|Ours|Marigold|
> |:---|:---:|:----:
> |Training Time (s/iteration) ($\downarrow$) | **0.28** | 1.08 |
> |Model Size (M) ($\downarrow$) | **96** | 860 |
> |GPU Memory (GB) ($\downarrow$)  | **10** | 30 |
>
> Our framework is a more efficient and effective solution compared with Marigold.
>
> ---
>
> We sincerely thank you once again for your valuable comments, which have greatly helped improve our work. We hope that the explanation above can well address your concerns.

---

> > ### Comment · Reviewer_8JiW · 2024-11-26
> > **Regarding to the rebuttal**
> >
> > Thank the author for giving a very detail rebuttal for my questions.
> > 1. The writing and the figure is now much clearer and more readable.
> > 2. Also, the ablate with or without self-improving really provide a clear advantage of this technique. Indeed I have misunderstood the Table 7, however author should rewrite it a bit to avoid such a confusion.
> > 3. For my question about "a variant without using diffusion-denoising process", I do not have a clear idea about this and I just wonder if it is possible to ablate this, (this remark is really minor and has little impact on my scoring).
> > 4. Superior performance compared to Marigold and StableNormal is really impressive so this is a plus. Thank the author for providing this.
> >
> > I have no further questions and have **increase the score to 6**

---

> > > ### Author Response · Authors · 2024-11-27
> > >
> > > We sincerely thank the reviewer for the reply! We are very glad that our reply has addressed your concern!
> > >
> > > We will continue to improve our work and incorporate your valuable advice into our paper to polish our presentation. If the reviewer has any additional follow-up questions, we are always happy to discuss!

---

### Official Review · Reviewer_dWWC · 2024-10-30

**Soundness:** 3
**Presentation:** 3
**Contribution:** 2
**Rating:** 6
**Confidence:** 4

**Summary:**

The paper introduces a framework that unifies generative and discriminative tasks within diffusion models. The framework enables simultaneous handling of multi-modal data generation and dense visual perception, by leveraging a diffusion-denoising approach to enhance both tasks. By creating synthetic data through a self-improving mechanism, the framework aims to boost the performance of dense visual perception tasks, particularly in low-data settings. Experimental results show improvements in state-of-the-art performance across various dense vision tasks, including semantic segmentation, depth estimation, and surface normal prediction.

**Strengths:**

1. This paper creates a unified diffusion model for unifying discriminative and generative tasks.
2. The writing of this paper is clear and easy to understand.

**Weaknesses:**

1. The proposed framework is limited to dense perception tasks such as segmentation, as the outputs are generated by diffusion models. Can it achieve other types of perception tasks such as categorization, instance segmentation, object detection, or text captioning?

2. The cycle of discriminative and generative learning in this framework only improves perception tasks but not generative tasks. I expect to see more discussions or experiments on how the framework can leverage multimodal information to improve RGB image generation.

3. Image generation performance has not been studied in the paper.

4. The self-imrpoving mechanism is one of the two major contributions of this work. I didn't see ablation study on using/not using it. It requires detailed ablation study on it to further reveal its effectiveness.

5. The performance improvement on semantic semgnetation and depth estimation is not significant compared to normal prediction. Could authors try to interpret this?

**Questions:**

In summary, the paper has the potential to be a qualified paper for ICLR. However, there are some major concerns as disccused above. If authors can address some of them, I will consider raising my rating.

---

> ### Author Response · Authors · 2024-11-25
> **Response to Reviewer dWWC**
>
> We sincerely appreciate the constructive feedback you have given for our work! Below, we provide the following clarifications to address your concerns:
>
> ---
>
> **W1: Extension of our approach for other types of tasks**
>
> Thank you for the comment! We would like to first kindly point out that our method is a general framework designed to accommodate different types of tasks by incorporating task-specific prediction heads. In this work, we focus on pixel-level dense prediction tasks, which are a series of primary and important tasks in computer vision. Nevertheless, following the suggestion, we conduct an additional evaluation on the multi-task benchmark of Tiny-Taskonomy, with categorization as one of the target tasks. The results are shown below:
>
>
> Tasks | Categorization | Segmentation | Depth | Normal
> |:---|:---:|:----:|:----:|:----:|
> | **Metrics** |**Top-1 Acc. (↑)** | **mIoU (↑)** | **RMSE (↓)** | **mErr (↓)**
> |TaskPrompter| 38.80 | 15.63 | 0.8350 | 28.87
> |Diff-2-in-1 on TaskPrompter (Ours)| **39.67** | **16.61** | **0.8289** | **28.35**
>
>
> The results validate that our framework can also provide improvement on other types of perception tasks beyond dense pixel prediction.
>
> ---
> **W2 & W3: Discussion on improving RGB image generation**
>
> Thanks for the comment! First, we would like to kindly point out that, in our default setting, the diffusion U-Net remains frozen, ensuring that the image generation capability remains consistent between our model and the backbone model (Stable Diffusion). More importantly, our framework extends this capability to enable ***multi-modal generation***, producing diverse and consistent image-annotation pairs --- a property not explored or achieved by previous approaches.
>
> Second, we have evaluated the generation performance both quantitatively and qualitatively. (1) In Section 5.4, we have assessed the quality and usefulness of the generated multi-modal data by measuring downstream task performances, as there are currently no direct metrics for evaluating such data. These results demonstrate the high quality of our generated paired data. (2) We have provided extensive visualizations of multi-modal generation results, as shown in Figures E-H in the appendix.
>
> Third, while our primary focus in this paper is on multi-modal generation, we appreciate the reviewer's insight that our framework holds the potential to leverage multi-modal information to enhance RGB generation. Conceptually, generating multi-modal visual attributes such as surface normals, depth, and semantic labels explicitly can provide more precise geometric and semantic information, which could significantly benefit RGB generation. We leave this as a very interesting direction for future exploration.
>
> We have revised the text in Lines 66-67, 130 to provide more clarification in the paper.
>
> ---
>
> **W4: Ablation study on the self-improving strategy**
>
> Thanks for the comment! We include the ablation study on using/not using the self-improving mechanism with Bae et al. on the NYUv2 surface normal benchmark as below:
>
>
> |Settings|11.25$^\circ(\uparrow)$|22.5$^\circ(\uparrow)$|30$^\circ(\uparrow)$|Mean ($\downarrow$)|Median ($\downarrow$)|RMSE ($\downarrow$)|
> |:---|:---:|:----:|:----:|:----:|:---:|:---:
> |Full Model | **67.4** | **83.4** | **88.2** | **13.2** | **6.5** | **22.0** |
> |w/o self-improving | 65.2 | 82.4 | 87.5 | 14.0 | 7.2 | 22.7 |
>
> We observe that the self-improving mechanism indeed further boosts the performance of the model by fostering an effective interaction between the discriminative and generative components in our framework.
>
> ---
> **W5: Performance improvement on semantic segmentation and depth estimation**
>
> For semantic segmentation, it is because the data regime between the settings of semantic segmentation and normal prediction is different. In the surface normal prediction task, there are only 795 available training samples as shown in Table 1. In comparison, for the semantic segmentation benchmark on ADE20K, there are about 20K training samples in the standard setting. The improvement of our method is more significant in a lower data regime.
>
> For depth estimation in Table 3, it is because this task is optimized together with other tasks in a multi-task setting. Therefore, the model also needs to balance and compromise among all the tasks, which makes the performance gain in Table 3 not as significant as the improvement in Table 1. However, when we place our scope of performance gain comparison to the same multi-task setting in Table 3, we can see that the performance gains of depth estimation and surface normal estimation within the single multi-task setting (Table 3) are on a comparable level.
>
> ---
>
> We sincerely thank you once again for your valuable comments, which have greatly helped improve our work. We hope that the explanation above can well address your concerns.

---

> > ### Comment · Reviewer_dWWC · 2024-11-25
> >
> > Thank you for the response. Some of my concerns have been addressed.
> >
> > However, I still have the concern for W2&W3. The authors mentioned that *"the diffusion U-Net remains frozen, ensuring that the image generation capability remains consistent between our model and the backbone model (Stable Diffusion)."* If the diffusion U-Net for image generation is frozen, this work would be more like applying a discriminative learning module (like VPD) to pre-trained diffusion, other than the claimed "2-in-1" framework. For a "2-in-1" framework, it is anticipated that both RGB and multimodal annotation can be improved due to the benefit from each other. This concern was also pointed out by reviewer LDAd.

---

> > > ### Author Response · Authors · 2024-11-27
> > >
> > > Thanks so much for your quick response! We would like to kindly clarify that the term "2-in-1" in our model name refers to **the single model handling both "multi-modal data generation" and "dense visual perception", as mentioned in Lines 16-18**. Therefore, what we are focusing on is the interplay between these two perspectives, as the term "2-in-1" suggests:
> > >
> > > - Generative learning: Multi-modal generation
> > > - Discriminative learning: Dense visual perception
> > >
> > > For generative learning, the evaluations in Section 5.4 and Figures E-H demonstrate the high-quality of our generated multi-modal data. However, as there is no standard way to direct quantify the multi-modal generated data, the focus of our paper is more leaning towards the evaluation on the discriminative performance. For the discriminative perspective, thanks to our integrated Diff-2-in-1 model that can simultaneously handle both dense perception and multi-modal generation, these two perspectives form a "positive feedback cycle" as shown in Figure 2 ($\theta_C$ provides synthetic data for $\theta_E$ to learn the discriminative perception, $\theta_E$ in turn provides guidance for $\theta_C$ to generate more useful multi-modal data, leading to the model converging in improved performance), which the previous approaches cannot achieve. In comparison, works like VPD only focus on discriminative learning, solely treating diffusion models as a feature extractor, and overlooking the unique diffusion-denoising process inherent in diffusion models. We made the unique exploration of the diffusion-denoising process within diffusion models to integrate both ***discriminative visual perception*** and ***multi-modal generation*** within our single Diff-2-in-1 model.
> > >
> > > Hope that our response addresses your concern! We are also glad to have further discussions!

---

> > > ### Author Response · Authors · 2024-12-01
> > >
> > > Dear Reviewer,
> > >
> > > Thanks again for your constructive comments! For your concern of whether RGB and multi-modal annotations can benefit from each other, we conduct the following experiments to show that **our joint modeling of discriminative and generative learning can indeed provide benefits for both tasks**. More specifically, we **unfreeze** the denoising U-Net in the diffusion model, and add a loss term on noise prediction for the RGB generation (the same generation loss term in Stable Diffusion) alongside the original discriminative loss. We find that the performance of both RGB generation and the discriminative task can be improved as follows:
> > >
> > > - For RGB generation, the FID of the synthetic RGB images for our self-improving stage improves from 48.53 (original Stable Diffusion checkpoint) to 40.40 (after our joint finetuning).
> > >
> > > - For the discriminative task on surface normal with Bae et al., the performance improvement is as follows:
> > > |Settings|11.25$^\circ(\uparrow)$|22.5$^\circ(\uparrow)$|30$^\circ(\uparrow)$|Mean ($\downarrow$)|Median ($\downarrow$)|RMSE ($\downarrow$)|
> > > |:---|:---:|:----:|:----:|:----:|:---:|:---:
> > > |Discriminative objective only (reported in the paper) | 67.4 | 83.4 |88.2 | 13.2 | 6.5 | 22.0 |
> > > |Generative and discriminative objectives (the new setting) | **68.6** | **84.4** | **89.1** | **12.8** | **6.4** | **21.6** |
> > >
> > > We can observe that, with the joint modeling of generative and discriminative training objectives and fine-tuning of the denoising U-Net, the quality of the generated RGB images gets improved, further boosting the performance of the discriminative perception task. As mentioned in the previous response, we froze the diffusion model in the original manuscript to prioritize building a more efficient and lightweight framework; but the experiments above show that with sufficient computational resources, fine-tuning the whole denoising U-Net can further improve the performance. We will include these new results into the next revision of our manuscript.
> > >
> > > We hope that our combined efforts  – (1) elaborating on our concept of an integrated framework, where a single model handles both "multi-modal data generation" (generative learning) and "dense visual perception" (discriminative learning), as clarified in our previous response, and (2) providing new experimental results in this response demonstrating that our framework improves both discrimination task performance and RGB image generation quality – fully address the reviewer’s remaining concern.
> > >
> > > Thank you once again for your insightful comments! We are looking forward to hearing from you, and are always glad to have further discussions!
> > >
> > > Best,
> > >
> > > Authors

---

> > > > ### Comment · Reviewer_dWWC · 2024-12-02
> > > >
> > > > Good job! The new results for joint RGB-annotation improvement address my remaining concern. I raise my rating.

---

> > > > > ### Author Response · Authors · 2024-12-02
> > > > >
> > > > > We sincerely thank the reviewer for the response! We are very glad that your concerns are all addressed!
> > > > >
> > > > > If the reviewer has any additional questions, we are always happy to discuss!
> > > > >
> > > > > Best,
> > > > >
> > > > > Authors

---

### Official Review · Reviewer_Ao2e · 2024-11-01

**Soundness:** 4
**Presentation:** 4
**Contribution:** 4
**Rating:** 8
**Confidence:** 4

**Summary:**

The authors present Diff-2-in-1, a unified framework that seamlessly integrates multimodal generation with discriminative dense visual perception using diffusion models. With a self-improving mechanism, multi-modal generation can be progressively enhanced in a self-directed manner. Extensive experiments show that Diff-2-in-1 achieves superior performance consistently across various discriminative backbones and generate high-quality multi-modal data.

**Strengths:**

1. The authors propose a novel method with self-improving mechanism to bridge high-fidelity image synthesis and dense visual perception tasks.
2. The paper is well-written and the structure is well-organized and easy to follow.
3. The authors conduct extensive experiments and ablation studies to demonstrate the superiority of their designs, providing impressive qualitative and quantitative results.

**Weaknesses:**

1. The content of preliminary can be appropriately reduced, and add several visual comparisons from the appendix to the main paper.
2. It can be observed from Table.2 that the Diff-2-in-1 only improve the diffusion-based the segmentation method by 0.5, which seems to be incremental. Please explain the reason.

**Questions:**

1. It can be observed that Taskprompter achieve better result for Saliency in Table.4. Why does the performance of Taskprompter decrease after adding Diff-2-to-1 to Taskprompter?

---

> ### Author Response · Authors · 2024-11-25
> **Response to Reviewer Ao2e**
>
> We are sincerely grateful for your positive feedback! We provide the following clarifications in response to your concerns:
>
> ---
>
> **W1: Reduce the content of preliminary and add visualizations**
>
> We thank the reviewer for the suggestion! We have adjusted the content of the preliminary and included more visualizations in the main paper in the revised manuscript accordingly, as shown in Lines 143-147 and Figure 5.
>
> ---
>
> **W2: Performance improvement in Table 2**
>
> We want to first kindly clarify that both DA-Fusion and our method are built upon the same baseline model (VPD). Therefore, the comparison between these two methods should be based on the performance gain over the baseline model instead of the absolute numbers. With this perspective, the performance gain from our method is 0.8 for the segmentation task, which is notably more significant than the performance gain of 0.3 brought by another data augmentation method DA-Fusion. As a reference, the performance gain of VPD over previous MAE/ConvNeXt-based methods is only 0.1 (from 53.6 to 53.7). Thus, we believe our improvement in the segmentation task is significant.
>
> ---
>
> **Q1: Decreased performance on saliency detection**
>
> Thanks for the great question! The reason is that, in multi-task learning where the model is optimized with multiple objectives, there are inherent trade-offs among all the tasks, especially when the optimization direction of some tasks conflicts with others, and the primary goal is to achieve the best overall performance across all tasks. In Table 4, we achieved improved performances for all the other tasks, and the overall performance of our method did get better, demonstrating the effectiveness of our method. However, if we want to improve the performance of a specific task under a multi-task setting, a simple yet effective way is to increase the weight of the objective corresponding to that task.
>
> ---
>
> We sincerely thank you once again for your valuable comments, which have greatly helped improve our work. We hope that the explanation above can well address your concerns!

---

> > ### Comment · Reviewer_Ao2e · 2024-12-02
> > **Response to authors**
> >
> > Thanks to the author for the response, the author's response addressed my concerns and I have no further questions. I will maintain my score.

---

> ### Author Response · Authors · 2024-12-02
>
> Dear reviewer,
>
> We sincerely thank you for the reply! We are glad that your concerns have been well addressed!
>
> Thank you once again for your constructive feedback!
>
> Best,
>
> Authors

---

### Official Review · Reviewer_Got7 · 2024-11-03

**Soundness:** 3
**Presentation:** 2
**Contribution:** 3
**Rating:** 6
**Confidence:** 5

**Summary:**

This paper introduces Diff2in1, a method to enhance discriminative capability by generating diverse and faithful synthetic data, particularly beneficial for discriminative learning with limited training data. The authors propose a self-improving learning mechanism that leverages synthetic data. Experiments and ablation studies demonstrate the effectiveness of the proposed method.

**Strengths:**

1. The approach of using synthetic data pairs and integrating a self-improving mechanism is interesting and valuable for discriminative tasks using pre-trained diffusion models.
2. The method shows promise in scenarios with limited training data.
3. Comprehensive experiments and ablation studies across multiple dense prediction tasks (surface normal estimation, semantic segmentation, depth estimation) and multiple datasets demonstrate the effectiveness and versatility of the approach.

**Weaknesses:**

1. The overall writing lacks clarity. The paper's main focus is on improving discriminative learning using synthetic data. Claiming it as "A single, unified diffusion-based model for both generative and discriminative learning" may be not proper, as the method does not improve generative learning.
2. The method section should clearly state which parameters are trained or frozen during different stages.
3. Figure 2 needs clarification. If understood correctly, in the left subfigure, the data creation corresponds to discriminative learning, with input and output corresponding to the generative learning in the right figure.
4. It's unclear whether the proposed method works with fully fine-tuned models for depth estimation, such as DepthCrafter and GeoWizard.
5. The synthetic data generated with SDEdit may only exploit data similar to the dataset, potentially limiting its ability to generate significantly different objects. It's unclear how the method improves discriminative capability on data with large domain shifts.

**Questions:**

See [Weaknesses] part

---

> ### Author Response · Authors · 2024-11-25
> **Response to Reviewer Got7**
>
> We are sincerely thankful for your insightful feedback on our work! We provide the following clarifications in response to your concerns.
>
> ---
>
> **W1: The overall writing lacks clarity. The paper's main focus is on improving discriminative learning using synthetic data. Claiming it as "A single, unified diffusion-based model for both generative and discriminative learning" may not be proper, as the method does not improve generative learning.**
>
> Thanks for the comment! We want to point out that improving discriminative learning is not our sole focus. We present a single integrated model for diffusion models to perform both discriminative perception and ***multi-modal*** generation, and we have also assessed the quality and usefulness of the generated multi-modal data in Section 5.4.
>
> Our work is significantly different from existing works like Marigold [a] or GeoWizard [b] which repurpose diffusion models for perception tasks. They only consider the optimization on the discriminative tasks by tuning the whole denoising U-Net. These approaches completely change the functionality of diffusion models from text-to-image generation to image-to-X (X is the predicted modality). Therefore, they lose the original capability of diffusion models to generate RGB images. In contrast, we still preserve the original generation capability of diffusion models and extend it to enable consistent multi-modal generation, including but not limited to RGB image synthesis.
>
> Additionally, we have revised Lines 66-67, 130 to provide further clarification in the updated manuscript.
>
> [a] Ke et al. Repurposing Diffusion-Based Image Generators for Monocular Depth Estimation. CVPR 2024.
>
> [b] Fu et al. GeoWizard: Unleashing the Diffusion Priors for 3D Geometry Estimation from a Single Image. ECCV 2024.
>
> ---
>
> **W2: Tunable parameters**
>
> Thanks for the comment! In our default settings, the pretrained latent diffusion remains frozen, with only the lightweight task head being tunable, therefore enabling more efficient and low-computation-budget tuning. Two exceptions are the VPD-based variant in Table 2, where finetuning U-Net is required by VPD, and the ablation of the finetuning strategy in Table B. We have revised related texts in Lines 322-323, 909-912 to clarify this.
>
> ---
>
> **W3: Clarification in Figure 2**
>
> We are sincerely grateful that the reviewer spotted this issue! There was indeed inconsistent notation in color illustration in the original version. We have updated Figure 2 in the manuscript.
>
> ---
>
> **W4: Extension of our method to fully fine-tuned models**
>
> We thank the reviewer for this insightful comment! As mentioned in our response to W1 above, one of the primary differences between our method and the recommended fully-tuned approaches is that those methods repurpose diffusion models for dense prediction by finetuning the denoising network, which makes them lose the capability of RGB generation. In comparison, our framework ***maintains the capability of RGB generation and further extends it to multi-modal generation***. This primary difference makes it difficult to directly instantiate our framework to the recommended models since their objective breaks the RGB generation. However, we do appreciate the suggestion and also believe that extending the general idea of bridging generation and dense perception to boost their performance is a promising future research direction.
>
> ---
>
> **W5: Boosting discriminative capacity on data with large domain shifts**
>
> This is a good point! We would like to first kindly point out that the diversity of the synthetic data generated with the SDEdit-style solution can be controlled by the hyperparameter $T$ which is the timestep of the diffusion process. Evidence from Figure 4 shows that a higher $T$ will lead to a more diverse generation of the multi-modal data while a smaller $T$ boosts the faithfulness of the generated data.
>
> Moreover, we conducted a cross-domain evaluation experiment in Table 6 showing that the out-of-distribution performance of our model is even better than the in-distribution performance of the baseline model that we build upon, which indicates that our framework provides improvement in situations with domain shifts.
>
> Finally, we want to point out that, naturally, it is challenging to fully solve the severe domain shift issue using a single source of training data. A more common solution is to train a model with multiple sources of data. In this case, the data generation strategy within our framework can further increase the diversity of the training data, thereby further boosting the robustness of our model on out-of-distribution test samples.
>
> ---
>
> We sincerely thank you once again for your valuable comments, which have greatly helped improve our work. We hope that the explanation above can well address your concerns!

---

### Official Review · Reviewer_LDAd · 2024-11-05

**Soundness:** 2
**Presentation:** 3
**Contribution:** 2
**Rating:** 6
**Confidence:** 5

**Summary:**

The paper presents Diff-2-in-1, a framework leveraging diffusion models for both multi-modal data generation and dense visual perception tasks. Unlike previous works that treat diffusion models as standalone components for perception or data augmentation, Diff-2-in-1 integrates generative and discriminative learning through a self-improving mechanism using EMA update. This mechanism involves two sets of parameters: one for data creation (θC) and another for data exploitation (θE). The training process iteratively enhances both generation and discriminative task performance with backpropagation and EMA update.

**Strengths:**

1.	Unified Framework: The paper integrates generative and discriminative processes to improve discriminative tasks using the diffusion model.
2.	Favorable performance on benchmarks: The method is evaluated on NYUD-MT and PASCAL-Context, and shows consistent performance improvements.

**Weaknesses:**

1.	The term “unified framework” is overstated and potentially misleading. The proposed method does not employ a single model to handle both data generation and prediction. Instead, it uses a pre-trained latent diffusion model for RGB image generation and a separate task-specific head for discriminative tasks. While there is interaction between these components during the update process, the term “unified” suggests a more cohesive integration than what is presented. From the description, it is likely to be expected that the model to perform both data generation and discriminative tasks by denoising process. Clarifying this would better align reader expectations with the actual implementation.
2.	Clarity. It is unclear whether θE and θC represent parameters of the task head alone or include latent diffusion parameters.
3.	Limited Novelty in Core Components: While the integration of generative and discriminative learning is an interesting direction, the core underlying techniques are adaptations of existing methods. The generation component relies on the widely used latent diffusion model, while the discriminative part uses VPD, a previously proposed method. The main technical contribution lies in the self-training mechanism using EMA updates, which is a general concept that could be applied to various ML tasks. The technical contribution and the motivation behind the problem are not well connected.
4.	Incomplete literature survey.
There have been recent works [1][2][3] on tackling discriminative tasks using the denoising process. These works are closely relevant but not mentioned nor discussed. These works should be discussed.
[1] Ke, Bingxin, et al. "Repurposing diffusion-based image generators for monocular depth estimation." Proceedings of the IEEE/CVF Conference on Computer Vision and Pattern Recognition. 2024.
[2] Liu, Xian, et al. "Hyperhuman: Hyper-realistic human generation with latent structural diffusion." arXiv preprint arXiv:2310.08579 (2023).
[3] Fu, Xiao, et al. "GeoWizard: Unleashing the Diffusion Priors for 3D Geometry Estimation from a Single Image." arXiv e-prints (2024): arXiv-2403.
5.	Unrelated works in the related work section. The related work section has two subsections. While the second subsection discusses relevant prior studies, the first subsection, titled “Pixel-level dense visual perception,” includes a list of works that are not relevant to the contributions of this paper.
6.	Minor Performance Gain: Performance gains over existing augmentation methods such as DA-Fusion are marginal.

**Questions:**

Questions
It is unclear whether θE and θC represent parameters of the task head alone or include latent diffusion parameters. Is the pre-trained latent diffusion fine-tuned during self-training, or does it remain frozen?

Comments
The paper is missing recent related works. While it explores the self-training of a framework using only VPD, there are more recent studies that apply the diffusion process to discriminative prediction tasks. Extending the idea to these diffusion-based discriminative methods would be an interesting direction.

---

> ### Author Response · Authors · 2024-11-25
> **Response to Reviewer LDAd [1/3]**
>
> We sincerely appreciate the detailed feedback you provided for our work! We provide the following clarifications in response to your concerns:
>
> ---
>
> **W1: Claim of the "unified framework"**
>
> We thank the reviewer for the suggestion! We would like to first kindly clarify that the generation aspect of our framework focuses on ***"multi-modal generation"*** rather than just RGB image generation, **for which we generate both RGB images and their corresponding pixel-wise annotations** (as highlighted in Lines 16-17, 50-52). Under this design, our model is a "unified" model with the task-specific head serving as a shared component used for both discriminative and generative tasks. Nevertheless, to avoid confusion, we have introduced additional clarifications (Lines 66-67, 130, etc.) and revised the manuscript (please refer to Lines 15, 48, 52, 74, etc.) changing the term "unified" to "integrated". We are happy to discuss and make further revisions.
>
> ---
>
> **W2 & Q1: Details of $\theta_E$ and $\theta_C$**
>
> Thanks for the comment! As indicated by Equations 3 and 4, $\theta_E$ and $\theta_C$ represent the entire parameters of our model. Notice that, ***not*** all parameters in $\theta_E$ / $\theta_C$ are required to be tuned for a more efficient and low-computation-budget finetuning.
>
> In our default experimental setting, we only finetuned the task head, except for the VPD-based variant (where we followed the full-weight finetuning strategy from VPD for a fair comparison) in Table 2 and the ablation of the finetuning strategy in Table B. However, whether or not finetuning the U-Net parameters does not affect the correctness of our self-improving mechanism in Equation 5, this equation always holds no matter whether we finetune the entire parameters or not. For our revised manuscript, we have revised the corresponding sections in Lines 322-323, and 909-912 for clarity.
>
> ---
>
> **W3: Novelty in Core Components**
>
> We would like to first kindly point out that the design of the core component is ***not*** the sole criterion for considering novelty. For example, VPD itself uses the standard design of plain FPN as its mask decoder; and both DIFT [a] and "A tale of two features" [b] even did not use any components except the pretrained diffusion models. We would like to further emphasize our key contributions as follows.
>
> - First, existing works, such as VPD and Marigold, utilize diffusion models solely for discriminative tasks with the loss of generation capability. As the reviewer acknowledges, "the integration of generative and discriminative learning is an interesting direction" – our proposed framework, which integrates both generation and discriminative perception, fills a gap in the current research landscape and offers a significant contribution for future work to continue exploring in this direction. **The novelty and promising exploration of the idea and our framework that integrates dense perception and multi-modal generation has also been acknowledged by Reviewers 8JiW, Ao2e, and Got7**.
> - Second, we want to emphasize that the generation task we focus on is the capability of "multi-modal generation", instead of sole RGB generation. Therefore, the fact that our model, which is trained only with discriminative objectives, also gains the capability of consistent multi-modal generation that can be used to boost the discriminative process is non-trivial and novel.
> - Finally, although EMA updates are broadly studied for settings like semi-supervised learning, which handles the case with unlabeled ***real data***, we further extend its application to the context of ***synthetic data utilization***. Specifically, we use EMA guidance from a model optimized for discriminative perception to update a model for multi-modal generation, while the multi-modal generation model, in turn, provides the synthetic data back to the model for discriminative perception. The underlying motivation is that while synthetic multi-modal data typically demonstrates high visual fidelity, its direct utility for discriminative learning remains uncertain. Therefore, we believe that this unique interaction between the generative and discriminative components, facilitated by the EMA mechanism, has seldom been studied and explored before.
>
> [a] Tang et al. Emergent Correspondence from Image Diffusion. NeurIPS 2023.
>
> [b] Zhang et al. A Tale of Two Features: Stable Diffusion Complements DINO for Zero-Shot Semantic Correspondence. NeurIPS 2023.

---

> ### Author Response · Authors · 2024-11-25
> **Response to Reviewer LDAd [2/3]**
>
> **W4: Missing recent related work**
>
> Thanks for the suggestion! We have added the discussion of these works in the related work section in Lines 120-126. At a high level, [1, 2, 3] take a different strategy of using diffusion models for discriminative learning – they ***repurpose*** diffusion models from text-to-image generation to dense prediction by finetuning the denoising U-Net. With such a design, they achieve promising results with the cost of totally losing the capability of generation. In comparison, we are exploiting the capability of diffusion models to discriminative perception, and at the same time, preserving the original RGB generation capability, and further expanding to multi-modal generation. Thanks to this unique design, we can utilize the multi-modal data to benefit discriminative learning, and further allow us to introduce a self-improving stage to enable interactions between generative and discriminative learning.
>
> We also provide a comparison with methods that repurpose diffusion models for discriminative tasks, empirically demonstrating that our framework is more flexible for such tasks. Below, we mainly focus on a comprehensive comparison with a representative method Marigold [1].
>
> - We compared three variants of Marigold on NYUv2 surface normal estimation in total. Variant 1, denoted as **Marigold (pretrain)**, is their released checkpoint that is trained on a mixed large dataset excluding NYUv2. For variant 2, we adopt the same setting as ours, using the 795 training samples to train Marigold from the Stable Diffusion checkpoint until convergence, denoted as **Marigold (SD)**. For variant 3, we further finetune their released checkpoint with 795 training samples from NYUv2, denoted as **Marigold (finetune)**. The results of the three variants, together with ours are reported below, as well as in Table F in the revised manuscript.
>
> |Settings|11.25$^\circ(\uparrow)$|22.5$^\circ(\uparrow)$|30$^\circ(\uparrow)$|Mean ($\downarrow$)|Median ($\downarrow$)|RMSE ($\downarrow$)|
> |:---|:---:|:----:|:----:|:----:|:---:|:---:
> |Marigold (pretrain) | 50.5 | 73.0 | 79.3 | 20.9 | 11.1 | 26.2|
> |Marigold (SD) | 48.8 | 76.8 | 84.0 | 18.1 | 11.5 | 25.8|
> |Marigold (finetune) | 64.0 | 82.4 | 87.8 | 14.2 | 7.7 | 22.3|
> |Diff2-in-1 on Bae et al. (Ours) | **67.4** | **83.4** | **88.2** | **13.2** | **6.5** | **22.0** |
>
> - All three variants lagged behind our model, indicating the effectiveness of our model design with the self-improving learning mechanism. Moreover, we observe that Marigold gets inferior performance when adapted to a specific domain with a limited amount of training data (795 samples). While finetuning from a well-trained model can help mitigate this issue, it still does not work as well as our proposed method. The reason is that tuning these diffusion-based perception models like Marigold, which require finetuning the denoising U-Net, is computationally expensive. In comparison, our approach only requires training a lightweight task head, which makes our framework more flexible and easier to train or fine-tune for new domains.
>
>
> - In addition, the comparison of the computational cost of our method and Marigold with a batch of images of shape (2, 512, 512, 3) is reported below:
>
> |Metrics|Ours|Marigold|
> |:---|:---:|:---:|
> |Training Time (s/it.) ($\downarrow$) | **0.28** | 1.08 |
> |Model Size (M) ($\downarrow$) | **96** | 860 |
> |GPU Memory (GB) ($\downarrow$)  | **10** | 30 |
>
> Our framework is a more efficient and effective solution compared with Marigold.
>
> ---
>
> **W5: Unrelated works**
>
> Thanks for the suggestions! We have removed this part and, as the reviewer suggests, replaced it with the discussion of more closely related works in the revised manuscript. Please refer to Section 2 in the revised manuscript for more details. In the original submission, we included a more general background for readers who may not be familiar with visual perception, because this paper focuses on dense visual perception.

---

> ### Author Response · Authors · 2024-11-25
> **Response to Reviewer LDAd [3/3]**
>
> **W6: Performance gain over DA-Fusion**
>
> We want to argue that the performance gain over DA-Fusion is ***significant*** across all the benchmarks shown in the paper.
>
> First, we want to clarify that the way to compare the performance gain between our method and DA-Fusion is not to compare the actual numbers of the two, but **the performance gain of these two models over the baseline models** (e.g. in Table 2, it is "the discrepancy between the last line and the third to last line" v.s. "the discrepancy between the second to last line and the third to last line")
>
> With this understanding, for surface normal prediction, the performance gain acquired by our method compared to DA-Fusion when iDisc as the base model is shown as follows:
>
> |Settings|11.25$^\circ(\uparrow)$|22.5$^\circ(\uparrow)$|30$^\circ(\uparrow)$|Mean ($\downarrow$)|Median ($\downarrow$)|RMSE ($\downarrow$)|
> |:---|:---:|:----:|:----:|:----:|:---:|:---:
> |Gain from DA-Fusion | 1.4 | 1.9 | 0.5 | 0.5 | 0.2 | 0.2|
> | Gain from Ours | **11.4** | **7.3** | **5.5** | **5.1** | **2.8** | **4.8** |
>
>
> It is clear to see that our performance gain compared with DA-fusion is significant.
>
> For semantic segmentation, our performance gain of 0.8 as shown in Table 2 is also obviously more significant than the gain of 0.3 brought by DA-Fusion. For reference, the performance gain of VPD compared with previous MAE/ConvNext-based methods is only 0.1 (from 53.6 to 53.7).
>
> For the multi-task benchmarks, the discrepancy between our method and DA-Fusion is also larger than the gap between DA-Fusion and the baselines in most metrics as shown in Tables 3 and 4.
>
> ---
>
> **Q2: Potential extension of this work to recent approaches**
>
> Thanks again for your insightful suggestions! As noted in our response to W4 above, while our work and the recent approaches suggested by the reviewer all leverage diffusion models to address visual perception tasks, they pursue different exploration directions and aim to achieve distinct objectives. These recent approaches like Marigold repurpose diffusion models for discriminative tasks by fine-tuning the entire diffusion U-Net, thereby changing the original function of diffusion models and **losing the capacity of RGB generation** during the fine-tuning process. In contrast, our method appends a lightweight task head while preserving the RGB generation capability – an essential property for enabling self-improvement within our framework. As a result, it is not straightforward to directly apply our approach to their architectures. Nevertheless, we sincerely appreciate the suggestion and believe that extending the general idea of bridging generation and dense perception to improve performance is a promising direction for future research.
>
> ---
>
> We sincerely thank you once again for your valuable comments, which have greatly helped improve our work. We hope that the explanation above can well address your concerns!

---

> ### Author Response · Authors · 2024-11-28
>
> Dear reviewer,
>
> We have posted our responses to your concerns and updated our manuscripts according to your suggestions, please check whether your concerns have been resolved and we are happy to have further discussions! Thanks again for your detailed and valuable review of our paper!
>
> Best,
>
> Authors

---

> ### Author Response · Authors · 2024-12-01
>
> Dear Reviewer,
>
> Thanks again for your detailed review! In our previous response, we have clarified and elaborated on our concept of an integrated framework, where a single model handles both "multi-modal data generation" (generative learning) and "dense visual perception" (discriminative learning). Here, we would like to present additional experiments demonstrating that **unfreezing the denoising U-Net in our framework and jointly learning both RGB generation and the discriminative objective enables mutual benefits**, enhancing both RGB generation and discriminative perception through the joint modeling.
>
> More specifically, we unfreeze the denoising U-Net in the diffusion model, and add a loss term on noise prediction for the RGB generation (the same generation loss term in Stable Diffusion) alongside the original discriminative loss. We find that the performance of both generative and discriminative tasks can be improved as follows:
>
> - The FID of the synthetic RGB images for our self-improving stage improves from 48.53 (original Stable Diffusion checkpoint) to 40.40 (after our joint finetuning).
> - The discriminative task on surface normal with Bae et al., the performance improvement is as follows:
> |Settings|11.25$^\circ(\uparrow)$|22.5$^\circ(\uparrow)$|30$^\circ(\uparrow)$|Mean ($\downarrow$)|Median ($\downarrow$)|RMSE ($\downarrow$)|
> |:---|:---:|:----:|:----:|:----:|:---:|:---:
> |Discriminative objective only (reported in the paper) | 67.4 | 83.4 |88.2 | 13.2 | 6.5 | 22.0 |
> |Generative and discriminative objectives (the new setting) | **68.6** | **84.4** | **89.1** | **12.8** | **6.4** | **21.6** |
>
> We can observe that, with the joint modeling of generative and discriminative training objectives, the quality of the generated images gets improved, further boosting the performance of the discriminative perception task. As mentioned in the previous response, we froze the diffusion model in the original manuscript to prioritize building a more efficient and lightweight framework; but the experiments above show that with sufficient computational resources, fine-tuning the whole denoising U-Net can further improve the performance. We will include these new results into the next revision of our manuscript.
>
> Thank you once again for your thoughtful comments! We are looking forward to hearing from you, and are always glad to have further discussions!
>
> Best,
>
> Authors

---

> ### Author Response · Authors · 2024-12-02
>
> Dear Reviewer LDAd,
>
> As the discussion period will come to an end in ***less than 24 hours***, we would like to send you a reminder about our responses as above to solve your concerns. Please check whether your concerns have been addressed. We are sincerely looking forward to hearing from you, and are always happy to have more discussions with you!
>
> Thank you once again for your insightful reviews!
>
> Best,
>
> Authors

---

### Author Response · Authors · 2024-11-25
**General Response**

We sincerely thank all reviewers for their detailed and constructive feedback. Reviewers recognize that our idea of bridging generative and discriminative learning is **novel and interesting** (Ao2e, 8JiW, Got7), our paper is **well-organized and easy to follow** (Ao2e, dWWC), and our method is **effective and robust** (LDAd, Got7, Ao2e, 8JiW), showing promise in scenarios with **limited training data** (Got7).

As a general clarification, we would like to emphasize that our framework addresses discriminative tasks while **maintaining the RGB generation** capability and further extending it to **multi-modal generation**. This is a crucial property that underpins our self-improving learning mechanism and distinguishes our approach from recent works suggested by the reviewers. These methods repurpose diffusion models for dense perception but **lose the capability of RGB generation** after fine-tuning the denoising U-Net.

We provide detailed responses to each reviewer to address all their questions. We hope that our responses can resolve your concerns. We have also revised the manuscript and included additional experimental results. A brief summary is provided below.

---

### **Summary of revision:**
Summary of the main revision of the manuscript  (marked in blue in the updated manuscript):

- Discussion of related works on tackling discriminative tasks using the denoising process. (Lines 120-126)
- Revision of Figure 2 to fix the inconsistent notation in color illustration. (Figure 2)
- Reduction on the preliminary and move some visualizations from the appendix to the main paper. (Lines 143-147, Figure 5)
- More clarified statement on the paper's main focus. (Lines 15, 48, 52, 66-67, 74, 130, etc., together with the text in Figures 1 and 2)
- Description of the frozen and trainable components in the framework. (Lines 322-323, 909-912)
- Add more clarification on the motivation of the section on the self-improving learning stage to make it more well-motivated. (Lines 264-265, 269, 311-314, 318-320)
- Add all the additional experimental evaluations in the appendix (Tables E, F, G, H)

---

### Author Response · Authors · 2024-12-04
**Summary of Author-Reviewer Discussion**

Dear Reviewers, ACs, SACs, and PCs,

We deeply appreciate your thoughtful feedback and constructive engagement, which helped to improve our submission! As the discussion phase comes to an end, we would like to summarize the key contributions of our work and highlight the additional analyses provided in the rebuttal, which have addressed the concerns of each reviewer. These clarifications, analyses, and results have been incorporated into the revised manuscript.

---

**Novel Contributions:**

- **Integrated Diffusion Model:** We introduce Diff-2-in-1, a novel, integrated diffusion-based model that bridges multi-modal data generation and dense perception, which is recognized by reviewers as an interesting and valuable direction to work on.

- **Self-improving Mechanism:** We design a self-improving mechanism to efficiently utilize the generated multi-modal data for discriminative learning, and the discriminative component, in turn, provides benefit for multi-modal generation, forming a "positive feedback loop."

- **Impressive Performance Improvement:** Our Diff-2-in-1 achieves consistent superior performance across multiple benchmarks compared with existing methods, including the recent diffusion-based perception methods like Marigold, with a more computationally efficient training cost.

---

**Key Insights from the Results in the Discussion Period:**

- **Benefits of the joint modeling:** By **unfreezing** the denoising U-Net, and adding a denoising loss term alongside the discriminative loss, the performance of **both RGB generation and the discriminative task** gets improved, verifying the effective design of our integrated diffusion model.

- **Impact of self-improving mechanism:** Ablation on the self-improving mechanism validated that our proposed **self-improving mechanism further evidently boosts the performance** of our integrated diffusion model.

- **Comparison with most recent approaches:** Extensive experiments show that our approach outperforms different variants of most recent diffusion-based approaches like Marigold, and is more **computational-efficient**.

- **Generalization to non-dense prediction tasks:** Additional experiment on Tiny-Taskonomy with *categorization* as one of the target tasks further verifies the general design of our framework, which is effective on both dense perception and global-level prediction tasks.

- **Clarity of the presentation:** We have updated the manuscript to address all specific suggestions regarding clarity and writing.

---

Thank you so much once again for your helpful comments and engagement in the discussion period!

Best,

Authors

---

### Meta-Review · Area_Chair_CSnD · 2024-12-18

**Metareview:**

The paper presents a diffusion-based framework designed to address multi-modal data generation and dense visual perception tasks. This approach leverages the diffusion-denoising process to enhance discriminative visual perception through the generation of multi-modal data that closely mimics the distribution of the original training set. It has been validated across a variety of discriminative backbones, demonstrating consistent performance improvements and high-quality multi-modal data generation.

***Strengths:***
- The framework successfully integrates generative and discriminative processes, optimizing the use of synthetic data for improved performance in discriminative tasks.
- Demonstrated effectiveness across multiple datasets and tasks, including NYUD-MT and PASCAL-Context, with consistent improvements over existing methods.
- Comprehensive experiments and ablation studies showcase the framework's versatility and effectiveness in enhancing discriminative learning with synthetic data.

***Weaknesses:***
There are shared concerns wrt the method clarity of this paper. Many details were initially unclear but have been improved post-rebuttal.
Lack of discussion and comparison with most recent literature. This one has also been improved after rebuttal.

The authors effectively addressed many of the concerns raised during the review process, enhancing the clarity and depth of their manuscript. Despite some initial shortcomings in clarity, the revised version and the authors' responses to the reviews have addressed these issues, leading to a better understanding and appreciation of the work's contributions. After careful consideration and discussion, we are delight to inform this paper is accepted.

**Additional Comments On Reviewer Discussion:**

All the main issues have been solved in the rebuttal, and many reviewers have raised their scores. Some reviewers have even changed their views from negative to positive about this paper. However, the authors still need to work on making the paper clearer.

There's also one more suggestion left from the discussion: the authors should mention *unfreezing the denoising U-Net in our framework and jointly learning both RGB generation and the discriminative objective enables mutual benefits*  in the paper.

---

### Decision · Program_Chairs · 2025-01-22

Accept (Poster)